# Data Augmentations for Improved (Large) Language Model Generalization

**Amir Feder** [*1,2], **Yoav Wald** [* 3], **Claudia Shi** [1], **Suchi Saria** [3] **and David Blei** [1]

[1] Columbia University, [2] Google Research, [3] Johns Hopkins University

## Abstract

The reliance of text classifiers on spurious correlations can lead to poor generalization at deployment, raising concerns about their use in safety-critical domains such as healthcare. In this work, we propose to use counterfactual data augmentation, guided by knowledge of the causal structure of the data, to simulate interventions on spurious features and to learn more robust text classifiers. We show that this strategy is appropriate in prediction problems where the label is spuriously correlated with an attribute. Under the assumptions of such problems, we discuss the favorable sample complexity of counterfactual data augmentation, compared to importance re-weighting. Pragmatically, we match examples using auxiliary data, based on diff-in-diff methodology, and use a large language model (LLM) to represent a conditional probability of text. Through extensive experimentation on learning caregiver-invariant predictors of clinical diagnoses from medical narratives and on semi-synthetic data, we demonstrate that our method for simulating interventions improves out-of-distribution (OOD) accuracy compared to baseline invariant learning algorithms.

## 1 Introduction

The reliance on spurious correlations is a significant challenge for Machine Learning (ML) safety as it can lead to performance degradation of deployed models. Spurious correlations are prevalent in various applications such as medical imaging [1, 2], text classification [3], and risk prediction systems [4]. Failures due to spurious correlations occur under distribution shift [5–7], which may result from differences in data recording protocols, shifts in the underlying population being monitored, or the way the ML tool is being used. In this paper, we focus on text classification and explore how using language models in a domain-informed way can help us avoid reliance on spurious correlations.

Consider a scenario where we want to make robust predictions about patients' conditions, probability of readmission, etc., using clinical narratives written in hospitals [8–10]. In this setting, a common issue arises due to clinical practice, where patients with certain conditions are directed to specific caregivers in the hospital. When we train a predictor from a single dataset that exhibits some correlation between caregiver-specific style and clinical outcomes, the predictor may unintentionally rely on the style to make predictions. This leads to poor generalization on unseen hospitals, i.e. failure to generalize out of distribution(OOD), due to changes in clinical practice [7]. However, collecting a dataset that is large enough to avoid such spurious associations is infeasible due to various reasons such as rare conditions, privacy concerns, etc. To tackle this problem, we propose leveraging available auxiliary data (e.g., time, document type, demographics) and incorporating knowledge about the causal structure of the problem to build a more robust classifier. For example, in the note classification task, we can use our knowledge that some auxiliary data, such as the patient's current state, can affect doctor assignment, to improve the classifier's robustness.

---

[*]Equal Contribution. Correspondence to amir.feder@columbia.edu

Causal inference often makes use of such auxiliary data and has now been used in a variety of ways to improve OOD generalization [6, 11–14]. Data augmentation methods have demonstrated impressive performance in these tasks as well [15–17], and with recent improvements in generative models, forming additional principles to incorporate domain knowledge into data augmentations seems like a promising path forward.

In this work we pursue this and develop *causally-driven data augmentation methods*, that leverage auxiliary data and domain knowledge. Intuitively, generating versions of clinical narratives as if they had been written by different caregivers, de-correlates the writing style from the patient condition we wish to predict. However, such data generation can be difficult to achieve in practice and problem-specific traits must be taken into account [18]. Observing that data augmentation can be treated as counterfactual outcome estimation under a causal formalism, motivates the use of causal inference methods that are commonly used for such tasks across the sciences. While our approach can be applied to many modalities of data, in this work we focus on text classification and harness the recent advances in LLMs towards counterfactual estimation. Our contributions are:

1. Through extensive experiments, we show how the use of language models in a manner that is informed by causal knowledge improves model robustness in challenging safety-critical tasks in healthcare. Furthermore, our findings are reinforced by experiments that incorporate semi-synthetic scenarios, and simulations where there are ground-truth counterfactuals.

2. We formalize counterfactual data augmentation in a prediction setting as a method to deconfound the target and a spuriously correlated attribute. We show how deconfounding improves OOD generalization. In a setting where sample complexities for alternative methods (re-weighting and invariance penalties) can be derived, we show favorable generalization bounds for accurately performed data-augmentation.

3. Our data-augmentation methods rely on common assumptions in the causal inference literature such as no unmeasured confounding and parallel trends in diff-in-diff [19], applied with LLMs. We believe that leveraging auxiliary data and assumptions about causal structure, along with the use of LLMs and other generative models, can be a fruitful framework for addressing many out-of-distribution generalization problems.

Next, we provide a brief survey of relevant work (§2). We then present a formal setting motivating counterfactual augmentation for OOD generalization (§3), our methods for counterfactual estimation and reason formally about the preferable sample complexity of our approach (§4). Finally, we present our main experimental results (§5) and discuss limitations and future directions (§6).

## 2   Related Work

**Invariant and Shift-stable Learning.** This paper contributes to the growing literature on invariant and shift-stable learning, which tackles the problem of learning models that generalizes across different distributions or settings. Invariant learning through feature pruning was pioneered by Peters et al. [11], and has since been developed for variable selection [12, 20] and representation learning [13, 21–26]. These methods have been applied in a range of domains, including natural science [11, 12, 20], causal estimation [27, 28], computer vision [13, 23], and NLP [29–32]. However, recent studies have highlighted limitations in many invariant learning approaches, particularly in achieving conditional independence [33–36]. Others have investigated learning of stable models by leveraging causal methods through techniques like graph-surgery [6, 14], that come with generalization guarantees. Yet others have explored the advantages of data augmentation [37, 38]. In this work, we combine the latter two approaches to improve OOD generalization for text based classification.

**Counterfactually Augmented Data.** To learn invariant predictors, a popular and straightforward approach is data augmentation. When data augmentation involves actions that go beyond simple manipulations (e.g. image rotations, crops etc.), it is often referred to as *counterfactual data augmentation* [37]. Constructing counterfactual instances that involve perturbations to confounding factors [39], or to the label [37, 38, 40], and incorporating them into the training data, breaks up correlations that we do not wish our model to exploit towards prediction. Most work on counterfactual data augmentation in text involve manual editing by humans, heuristic keyword replacement, or automated text rewriting [37, 39, 41–50]. Manual editing is accurate and effective [38, 51] but expensive, hence our goal is to *make counterfactual data augmentation scalable*, demanding smaller

human effort. Keyword-based methods can be limited in coverage and difficult to generalize across languages [52]. Generative approaches offer a balance of fluency and coverage [53], but generating meaningful counterfactuals is challenging [54]. Our work departs from previous techniques by using *causal auxiliary data structure and LLMs* to alleviate this challenge and generate plausible counterfactual data augmentations.

**Clinical Notes.** Clinical notes are the backbone of electronic health records, often containing vital information not observed in other structured data Kreimeyer et al. [55]. Clinical NLP involves identifying this information, and standardized datasets and competitions exist for this purpose [56–60]. Best performing approaches have leveraged transformer architectures both for token-level classification tasks [61–64], and for using complete clinical records [65, 66]. Recently, large language models (LLMs), similar to those we use to generate counterfactual notes, were shown to have clear potential for improving clinical NLP systems [67, 68]. In our experiments, we follow recent papers in clinical NLP addressing challenges of degraded performance across different hospitals [69–71].

## 3 Problem Setting

To formally analyze how counterfactual data augmentation helps OOD generalization, we consider a setting where the label is spuriously correlated with a known attribute. This setting has been used previously to study learning with "shortcuts" [25] and spurious correlations [29]. We note that our approach is applicable and valid under additional settings and causal graphs (e.g. "purely spurious" problems defined in Wang and Veitch [72]) and we elaborate on this at **??**. The data generating process used here motivates counterfactual data augmentation in a principled manner, as it describes the main problem we study and it is possible to analytically compare sample complexity with an alternative solution (see section 4.3).

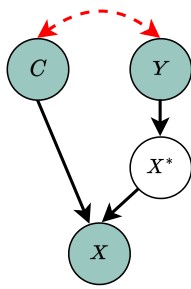

Consider a classification problem with $L$ classes, where the label $Y$ is correlated with a certain attribute $C$ in the training data and this correlation may change arbitrarily at test time (denoted by a red edge $C \leftrightarrow Y$ in fig. 1). In our medical notes example, $C$ is the caregiver writing the note and $Y$ is the underlying condition we wish to diagnose. We denote the number of caregivers in our training data by $[K]$. For a given loss function $\ell : \mathbb{R}^L \times [L] \to \mathbb{R}$ and distribution $P$, we denote the expected loss of a hypothesis $h : \mathcal{X} \to \mathbb{R}^L$ by $\mathcal{R}_P^\ell(h)$ and its expected accuracy by $\mathcal{R}_P^{\ell_{01}}(h)$. The data-generating process is depicted by the causal model in fig. 1, for our motivating example of clinical notes classification $X$ is a vector representation of the clinical note and $X^*$ is an unobserved sufficient statistic, representing all the relevant information about $Y$ in the note that is unaffected by the writing style of the caregiver. Let us formally define this setting.

Figure 1: Prediction problem with a spuriously correlated attribute.

**Definition 1.** *We denote the set of distributions induced by interventions on a causal model with the structure in fig. 1 by*

$$\mathcal{P} = \left\{ P(X \mid X^*, C) P(X^* \mid Y) P(Y) \tilde{P}(C \mid Y) \ : \ \tilde{P}(C \mid Y = y) \in \Delta^{K-1} \ \forall y \in [L] \right\},$$

*where all distributions other than $\tilde{P}(C \mid Y)$ are fixed. In a prediction problem with a spuriously correlated attribute, the learner is provided with a set $\{(\mathbf{x}_i, y_i, c_i)\}_{i=1}^N$ sampled i.i.d from $P_{train} \in \mathcal{P}$. We assume that $X^* = e(X)$ almost surely for some $e : \mathbb{R}^d \to \mathbb{R}^{d^*}$.*

In this problem, once $X^*$ is recovered no additional information from $X$ is needed to predict $Y$. We can also see from the graph that interventions on $\tilde{P}(C \mid Y)$ do not change the conditional distribution $P(Y \mid X^*)$. Therefore an optimal solution that does not rely on $C$ is $h^*(\mathbf{x}) = \arg\max_{y \in L} P(Y = y \mid e(\mathbf{x}))$. In clinical note classification, $X^*$ represents all the information in the note about the patient conditions, unsullied by the writing style of caretaker $C$. To obtain $h^*(\mathbf{x})$ we will rely on risk minimization w.r.t a distribution where $Y$ and $C$ are uncorrelated.

## 3.1 Learning Robust Classifiers when Counterfactuals are Available

Consider the unconfounded distribution $P_\perp \in \mathcal{P}$ that is given by intervening on $C$, setting it independent of $Y$ and uniformly distributed, $\tilde{P}(C \mid Y) = P_{\text{unif}}(C)$. An optimal classifier under $P_\perp$ has the following min-max optimality guarantee. [2]

**Lemma 1.** *For the prediction problem in definition 1, the Bayes optimal classifier under the unconfounded distribution $P_\perp \in \mathcal{P}$ where $C$ is uniformly distributed and independent of $Y$ is $h^*(\mathbf{x}) = \arg\max_{y \in [K]} P_\perp(Y = y \mid X^* = e(\mathbf{x}))$. It is a minimizer of $\min_{h:\mathcal{X} \to [L]} \max_{P \in \mathcal{P}} \mathcal{R}_P^{\ell_{01}}(h)$ and $\mathcal{R}_P^{\ell_{01}}(h^*) = \mathcal{R}_{P_\perp}^{\ell_{01}}(h^*)$ for all $P \in \mathcal{P}$.*

Hence we would like to minimize risk w.r.t $P_\perp$ and we cannot do that directly by via ERM since our training data is sampled from $P_{\text{train}} \neq P_\perp$. Instead we consider risk minimization over an augmented dataset that contains counterfactual instantiations of our training data under different values of $C$.

**Minimizing $\mathcal{R}_{P_\perp}$ via Counterfactual Data Augmentation.** Returning to our motivating example, assume that we could generate clinical notes for all alternative scenarios. That is, obtain the clinical notes that would have been written if each patient had been seen by all possible caregivers $c \in [K]$ and each caregiver had written their own version of the note $\mathbf{x}_i(c)$. Given these counterfactual clinical notes, we seek a hypothesis that minimizes the average loss over all such possible scenarios, denoted by $\widehat{\mathcal{R}}_{\text{aug}}^\ell(h)$.

**Definition 2.** *Consider a prediction problem with a spuriously-correlated attribute (see Definition 1). For a given example $\mathbf{x}_i$, we denote its counterfactual with attribute value $c \in [K]$ as derived from the corresponding causal model, by $\mathbf{x}_i(c)$. For estimates of the counterfactuals $\{\hat{\mathbf{x}}_i(c)\}_{i \in [N], c \in [K]}$ and a hypothesis $h \in \mathcal{H}$, the counterfactually augmented empirical risk is*

$$\widehat{\mathcal{R}}_{aug}^\ell(h) = \frac{1}{NK} \sum_{i \in [N], c \in [K]} \ell\left(h\left(\hat{\mathbf{x}}_i(c)\right), y_i\right). \tag{1}$$

We use approximate counterfactuals $\hat{\mathbf{x}}_i(c)$ in our definition to highlight that in practice we cannot obtain a precise estimate of $\mathbf{x}_i(c)$. In the ideal case where $\hat{\mathbf{x}}_i(x) = \mathbf{x}_i(c)$, the expected loss $\mathcal{R}_{\text{aug}}^\ell(h)$ where $N \to \infty$, satisfies $\mathcal{R}_{\text{aug}}^\ell(h) = \mathcal{R}_{P_\perp}^\ell(h)$. This follows by a simple derivation and it is part of a claim we give later in Lemma 2. Hence obtaining this dataset is useful for our goal of minimizing risk under $P_\perp$. Our main challenge is then to derive effective approximations for counterfactuals such as clinical notes under alternative writing styles.

## 4 Assumptions and Algorithms for Estimating Counterfactuals

Perfectly capturing writing style is a strong assumption. Even if we could perfectly model writing styles, we only observe a limited set of variables - the actual notes $x$, outcomes $y$, and assigned caregivers $c$. We do not observe all factors that could influence what each caregiver would write. To alleviate this problem, we make use of auxiliary data $M$ that is available during training, but might not be available in deployment.

As an example, consider two caregivers $c$ and $\tilde{c}$, where a note $\mathbf{x}_i$ was written by $c_i = \tilde{c}$. We want to estimate what $\mathbf{x}_i(c)$, the note caregiver $c$ would have written, might look like. To this end we will build a model $\tau_c(\cdot)$ that takes data and generates a note in caregiver $c$'s style. Now suppose caregiver $c$ usually sees patients with high blood pressure and always includes blood pressure values in notes, while $\tilde{c}$ rarely does. A naive model estimating $\hat{\mathbf{x}}_i(c) = \tau_c(\mathbf{x}_i)$ based only on $c$'s notes may fill in false blood pressure information, conflating that with $c$'s style. Including vitals data like blood pressure, typically recorded in a patient's health record, can provide additional context for our model. This extra information can assist the model in reasoning about external/background variables, leading to more accurate estimates.

---

[2]This claim is shown in Makar et al. [25], appendix A includes a proof for completeness. We set the distribution over $C$ in $P_\perp$ as uniform for simplicity, the derivation for non-uniform distributions is analogous.

## 4.1 Identification of the Counterfactual Distributions

To make effective use of this data, we suggest that the input to the model $\tau_c : \mathcal{X} \times \mathcal{M} \to \mathcal{X}$ will include a baseline text to be edited and auxiliary data $\mathbf{m}$. Intuitively, accounting for confounding between the identity of the caregiver $C$ and the text $X$, with auxiliary data $M$ should result in improved augmentation.

We formalize this intuition using an assumption from causal inference. To identify the counterfactual text distributions using the observed distribution, we assume strong ignorability [73–75]

**Assumption 1** (Strong ignorability). *For all $P \in \mathcal{P}$ it holds that $X(c) \perp\!\!\!\perp C \mid M$, and for all values of $\mathbf{m} \in \mathcal{M}$, $P(\mathbf{m}) > 0$.*

Under this assumption, we can rewrite the counterfactual distribution with the observed distribution,

$$P(X(c)) = \int P(X(c) \mid M = \mathbf{m})P(M = \mathbf{m})d\mathbf{m} = \int P(X \mid C = c, M = \mathbf{m})P(M = \mathbf{m})d\mathbf{m}.$$

However, in practice, we do not observe many samples from $P(X \mid C = c, M = \mathbf{m})$, making it a poor approximation for the counterfactual distribution. We address this by using counterfactual data augmentation [37]. Formally, we assume that for all possible counterfactual distributions $c \in [K]$, there exist a function $\tau_c$ that maps from the observed distribution $P(X \mid M = \mathbf{m})$ to the target counterfactual distribution $P(X(c) \mid M = m)$.

We approximate the loss under the counterfactual distributions through the empirical loss produced by data augmentation. That is, for a hypothesis $h \in \mathcal{H}$

$$\mathbb{E}_{P(X(c))}[\ell(h(\mathbf{x}), y)] \approx \frac{1}{N} \sum_{i \in [N]} \tau_c(\mathbf{x}_i, \mathbf{m}_i).$$

Note that whenever the text in the training set is already written by caregiver $c$, i.e. $c_i = c$, we will simply keep the original text $\mathbf{x}_i$

**Evaluation of Augmented Distribution.** The right hand-side of the above equation is a Monte-Carlo estimator of the distribution of augmented notes, which averages the distributions $\tau_{*,c}(P_{\text{train}}(X, M))$ over all caregivers $c \in [K]$. The distribution $\tau_{*,c}(P_{\text{train}}(X, M))$ is aimed to follow the style of caregiver $c$. While the observed samples from one counterfactual distribution may not be sufficient to approximate the whole distribution, they can be used to assess the quality of the counterfactual augmentation algorithm $\tau_c$.

High-quality counterfactual estimation, as measured by small distributional divergence between our estimator and the target distribution, will help in lowering the upper bound on the risk $\mathcal{R}^\ell_{P_\perp}(h)$ (see lemma 2 in section 4.3). Then to estimate divergences between these two distributions, we may use validation sets from our training data. A sample from $\tau_{*,c}(P_{\text{train}}(X, M))$ is obtained simply by running training data through $\tau_c$, while a sample from $P(X(c))$ can be obtained either by adjusting for $M$, or we can obtain a sample from $P(X \mid C = c, M = \mathbf{m})$ for each value of $\mathbf{m}$ and compare that to a sample obtained by augmenting validation data where $M = \mathbf{m}$. In both cases two-sample tests can be applied and obtain estimates of divergences between the two distributions. That is of course as long as positivity holds, i.e. the second part of the assumption, as otherwise we will not be able to obtain samples of $P(X \mid C = c, M = \mathbf{m})$ for certain values of $\mathbf{m}$ and $c$.

We now describe the estimation methods that obtain $\tau_c$. The methods are based on classical causal inference methods, applied to our high-dimensional setting, and relying on the auxiliary data $M$.

## 4.2 Methods for Estimation of Counterfactuals

Counterfactual estimation is an established problem in causal effect estimation [74, 76, 77]. Here we adapt identification strategies and estimation procedures in the causal literature to estimate $\mathbf{x}_i(c)$. Our framework for estimating counterfactuals *CATO* (**C**ausal-structure Driven **A**ugmentations for **T**ext **O**OD Generalization) involves the use of an LLM to model the conditional probability distribution of text. Counterfactuals are formed by matching similar auxiliary data examples or manipulating texts' vector representations, as described below.

**Prompting with matched examples.** Our first estimation method in Algorithm 1(B) draws insights from matching [76]. We construct a prompt for an LLM, that given an original text $\mathbf{x}$ and a set of

| **Algorithm 1** *CATO* |
| --- |

**Input:** Training set $\{(\mathbf{x}_i, y_i, c_i, \mathbf{m}_i)\}_{i=1}^N$
   Hypothesis class $\mathcal{H}$
   Version $\in \{(A), (B)\}$
   **Optional** pre-treatment data $\{(\mathbf{x}_{\mathrm{pre},i})\}_{i=1}^N$
**Output:** A hypothesis $h_{\mathrm{aug}}(\mathbf{x})$
  1: **if** Version = $(A)$ **then**
  2:     Get $\tau_c(\mathbf{m}, \mathbf{x})$ with preprocess (A)
  3:     Get $\hat{\mathbf{x}}_i(c) = \tau_c(\mathbf{x}_{i,\mathrm{pre}}, \mathbf{m}_i)\ \forall i \in [N]$
  4: **else**
  5:     Get $\tau_c(\mathbf{m}, \mathbf{x})$ with preprocess (B)
  6:     Get $\hat{\mathbf{x}}_i(c) = \tau_c(\mathbf{x}_i, \mathbf{m}_i)\ \forall i \in [N]$
  7: **end if**
  8: **return** $h_{\mathrm{aug}} \in \mathcal{H}$ that minimizes $\widehat{\mathcal{R}}_{\mathrm{aug}}^{\ell}$.

| **Pre-process** *CATO* **(A)** |
| --- |

**Assume:** $\mathbf{m}$ includes the label $y$ and pre-treatment attribute $c_{\mathrm{pre}}$, among other auxiliary data. We are given $\{\mathbf{x}_{j,\mathrm{pre}}\}_{j=1}^N$.
  1: Set $\rho(c_j, \mathbf{m}_j) = \mathbf{x}_j - \mathbf{x}_{j,\mathrm{pre}}$ for $j \in [N]$.
  2: **return** $\tau_c(\mathbf{x}, \mathbf{m}) := \mathbf{x}_{\mathrm{pre}} + \rho(c, \mathbf{m})$

| **Pre-process** *CATO* **(B)** |
| --- |

**Assume:** $\mathbf{m}$ includes the label $y$ among other auxiliary data.
  1: **return** prompt $\tau_c(\mathbf{x}, \mathbf{m})$ that rewrites $\mathbf{x}$ in the style of matching examples with attribute $c$, i.e. $\{\mathbf{x}_j : (\mathbf{m}_j, c_j) = (\mathbf{m}, c)\}$.

context notes, asks the LLM to rewrite $\mathbf{x}$ in their style. Now given text $\mathbf{x}$ with auxiliary data $\mathbf{m}$ that we wish to estimate with counterfactual value $c$ (i.e. writing style), $\tau_c(\mathbf{x}, \mathbf{m})$ runs this prompt with context notes whose auxiliary data is similar to $\mathbf{m}$ and their attribute value equals the desired $c$.

**Diff-in-diff estimation.** The procedure we use for medical note generation relies on additional structure involving panel data (i.e. data collected over time intervals across several individuals). In our case of clinical narratives, a narrative is usually consisted of several notes taken over the course of a patient's visit and each may be written by a different caregiver. Prediction is made using the release note from the hospital whose embedding consists our features $\mathbf{x}$. For simplicity let us consider a single note $\mathbf{x}_{\mathrm{pre}}$ taken prior to $\mathbf{x}$. Difference-in-difference [19, 78, 79] estimation of causal effect is based on the parallel-trends, or constant effect assumption that two units $i, j$ with similar pre-treatment conditions would have seen the same effect had they been given the same treatment. In our case, the treatment is an assignment to a certain caregiver. Hence we assume our auxiliary data $\mathbf{m}$ includes $c_{\mathrm{pre}}$, the caregiver assigned pre-treatment.

**Assumption 2** (constant effect). *Let $\mathbf{x}_{i,pre}$ be the pre-treatment features for unit $i$, and assume $\mathbf{m}_i$ includes the pre-treatment attribute $c_{i,pre}$. There exists a function $\rho : [K] \times \mathcal{M} \to \mathcal{X}$ such that $\mathbf{x}_i(c) = \mathbf{x}_{i,pre} + \rho(c, \mathbf{m}_i)$.*

Under this assumption, to calculate $\mathbf{x}_i(c)$ we can use any unit $j$ for which $\mathbf{m}_i = \mathbf{m}_j$ and has $c_j = c$ to estimate $\rho(c, \mathbf{m}_i) = \mathbf{x}_j - \mathbf{x}_{\mathrm{pre},j}$. The resulting estimation procedure is given in algorithm 1(B) and illustrated in section 4.2.

Before empirically evaluating our methods, we discuss alternatives for learning robust classifiers in our setting, and how their properties fair compared to counterfactual augmentation.

### 4.3 Why Bother with Counterfactual Data Augmentation?

Reasoning about counterfactuals with problem-specific domain knowledge is a considerable challenge, and it is interesting to see whether this has any advantage in learning robust classifiers compared to methods that rely on less stringent assumptions. A simple alternative to approximating counterfactuals involves re-weighting the loss function (see e.g. Makar et al. [25], Shimodaira [80]).

**Reweighting baseline.** Intuitively, re-weighting samples from the uncorrelated distribution $P(Y, C) = P(Y)P(C)$ by setting for each example $i$ a weight $w_i = P_{\mathrm{train}}(Y = y_i)P_{\mathrm{train}}(C = c_i)/P_{\mathrm{train}}(Y = y_i, C = c_i)$ and

| Time \ Patient | $T-1$ (Progress) | | $T$ (Discharge) | |
| --- | --- | --- | --- | --- |
| | Caretaker | Note | Caretaker | Note |
| $i$ | | $x_{i,pre}$ | | $x_i$ |
| $j$ | | $x_{j,pre}$ | | $x_j$ |

Panel A: Matching patients using auxiliary data

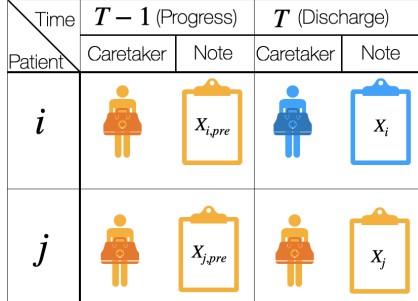

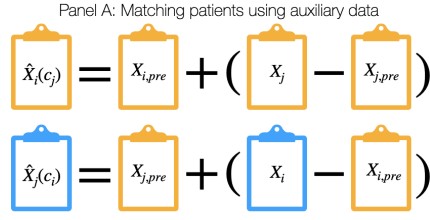

Panel B: Generating counterfactual discharge summaries

Figure 2: Generating counterfactual clinical notes for patients using auxiliary data with Algorithm 1(A).

minimizing the weighted empirical risk:

$$\hat{\mathcal{R}}_{\mathbf{w}}^{\ell}(h) = \frac{1}{m} \sum_{i \in [m]} w_i \ell(h(\mathbf{x}_i), y_i).$$

It can be proved that at the limit of infinite data the method learns a min-max optimal hypothesis, as it also effectively minimizes $\mathcal{R}_{P_\perp}^{l}$ (see [25]). While augmentations may not seem advantageous for identifying the correct hypothesis, reweighting can demand a larger sample to identify the correct hypothesis, particularly when $Y$ and $C$ are highly correlated.[3]

**Comparing sample complexities.** To make this statement precise, we can apply the bounds from Cortes et al. [81] and compare them with an upper bound that we will derive for our method in Lemma 2. To this end, let us consider the exponent of the Rényi divergence as a measure of dependence between $Y$ and $C$ in the training data. The divergence is given by $d_{\alpha,\text{train}}(Y,C) = [\sum_{y \in [L], c \in [K]} P_{\text{train}}^{\alpha}(Y = y, C = c)/P_{\text{train}}^{\alpha-1}(Y = y)P_{\text{train}}^{\alpha-1}(C = c)]^{\frac{1}{\alpha-1}}$, and we may derive the following bound for a hypothesis $h \in \mathcal{H}$ and any $\delta \in [0,1]$:

$$\mathcal{R}_{P_\perp}^{\ell}(h) \leq \hat{\mathcal{R}}_{\mathbf{w}}^{\ell}(h) + \sqrt{\frac{2d_{2,\text{train}}(Y,C) \cdot \log(1/\delta)}{N}} + \frac{d_{\infty,\text{train}}(Y,C) \cdot \log(1/\delta)}{N}. \tag{2}$$

A complementary lower bound on $\hat{\mathcal{R}}_{\mathbf{w}}^{l}(h)$ can also be derived based on results in Cortes et al. [81]. To compare this with counterfactual augmentations, denote our augmentation model by $\tau : \mathcal{X} \times \mathcal{M} \to \mathcal{X}^K$, which is some measurable function whose output's $c$-th coordinate is the counterfactual estimate w.r.t. caregiver $c$, i.e. $\hat{\mathbf{x}}(c) = \tau_c(\mathbf{x}, \mathbf{m})$. The following statement quantifies the relation between the accuracy of $\tau(\cdot)$ in approximating counterfactuals and the classification accuracy of a model learned from the augmented data, via minimization of $\hat{\mathcal{R}}_{\text{aug}}^{\ell}(h)$ in eq. (1).

**Lemma 2.** *Consider a prediction problem with a spuriously-correlated attribute (definition 1), a measurable function $\tau : \mathcal{X} \times \mathcal{M} \to \mathcal{X}^K$, and let $d_1(P,Q)$ denote the total variation distance between two distributions $P, Q$. Further let $h^*, h_{aug}^*$ denote the optimal hypotheses w.r.t $\mathcal{R}_{P_\perp}^{\ell_{01}}, \mathcal{R}_{aug}^{\ell_{01}}$ respectively and let $\lambda_{aug} = [R_{P_\perp}^{\ell_{01}}(h_{aug}^*) - R_{P_\perp}^{\ell_{01}}(h^*)]$. For any hypothesis $h \in \mathcal{H}$, and any $\delta \in (0,1)$ it holds that with probability at least $1 - \delta$ over the draw of the training set,*

$$\mathcal{R}_{P_\perp}^{\ell_{01}}(h) \leq \hat{\mathcal{R}}_{aug}^{\ell_{01}}(h) + \sqrt{\frac{\log(1/\delta)}{N}} + K^{-1} \cdot \sum_{c \in [K]} d_1(\tau_{c,*}(P_{train}(X,M)), P(X(c))) + \lambda_{aug}.$$

The divergence $d_1(\tau_{c,*}(P_{\text{train}}(X,M)), P(X(c)))$ is a distance between the true distribution over counterfactual instances $P(X(c))$ and our augmented data $\tau_{c,*}(P_{\text{train}}(X,M))$.[4] Divergences other than total-variation can be used, resulting in tighter bounds, e.g. see Ben-David et al. [82]. As we generate better counterfactuals this divergence decreases, and it can also be shown that $h^*$ and $h_{\text{aug}}^*$ coincide. Hence $\lambda_{\text{aug}}$ vanishes and the bound scales with $N^{-\frac{1}{2}}$, resulting in a gain of factor $d_{2,\text{train}}(Y,C)$ over the upper bound on $\hat{\mathcal{R}}_{\mathbf{w}}^{\ell_{01}}(h)$ in Equation (2). We discuss the details in the appendix, and in Section 5 we show this empirically through simulations.

**Takeaways and additional baselines.** We emphasize that that the counterfactual datapoints should not be interpreted as "more data" in the sense of i.i.d training examples, they rather embody knowledge about how the causal mechanism that generates features $X$ acts under interventions on the attribute $C$ (as formalized in e.g. [74, 83]). This translates into an improved sample complexity towards risk minimization on $P_\perp$. Counterfactuals are not the only type of causal knowledge that may be leveraged for learning more stable models. Many data dependent penalty terms have been proposed to impose conditional independence constraints drawn from the causal structure of the problem. Theory on these methods usually shows improved OOD performance under infinite data [13, 22, 24, 29]. Our baselines include a method based on the Maximum-Mean Discrepancy (MMD) from Makar et al. [25] who show improved sample complexity under a linear hypothesis class.

---

[3]We remark that other works discuss the potential benefits of data augmentation for identification in other problem settings, e.g. [72, Thm. 9] and [17].

[4]The notation $\tau_{c,*}(\cdot)$ denotes the pushforward measure. We note that in our implementation $\tau_c$ is data dependent and we ignore this dependence to enable a simple analysis.

# 5 Experiments

We empirically study the following questions: (1) Can *CATO* enhance OOD performance of downstream classifiers? (2) Does it surpass the combination of reweighting and invariance penalties? (3) Is it more effective than alternative augmentation techniques, thus demonstrating the usefulness of the causal graph? (4) How sensitive is *CATO* to quality of counterfactuals?

These questions seek to establish causally-motivated augmentations as a practical approach for improving OOD performance. We address Q#1,#2 and #3 through our theoretical foundation and across all empirical studies, while Q#4 is explored in the synthetic experiments. Further details about the experimental setup, including data statistics, model hyperparameters, and data splits, can be found in Appendix B. Table 1 provides an overview of the tasks we experiment with.

| Input ($x$) | Label ($y$) | ID Data | OOD Data | Spurious Feature ($c$) | auxiliary data ($m$) |
|---|---|---|---|---|---|
| Clinical Narratives | Condition Prediction Note Segmentation Demographic Traits | MIMIC-III | i2b2-2010 partner data i2b2-2006 | Caregiver ID | Medications, Lab Results, Vitals |
| Restaurant Reviews | Restaurant Rating | CEBaB | CeBAB-Spurious | Food-mention | Service, Noise, Ambiance, Food |
| Synthetic Data | $\{0, 1\}$ | Gaussians | | $\{0, \cdots, 7\}$ | – |

Table 1: Description of all our tasks and their corresponding experimental setup.

**Baselines.** We compare *CATO* to several baselines:

- Observational - Baseline model trained on the original data. *PubMED BERT* [84] for *clinical narratives*, logistic regression for the *restaurant reviews* and *synthetic* experiments. [5]
- Reweighting - Baseline model with sample reweighting as in Makar et al. [25].
- MMD - Baseline model with an MMD penalty as in Makar et al. [25], Veitch et al. [29].
- IRM - Baseline model with the IRMv1 penalty as in Arjovsky et al. [13].
- GroupDRO - Baseline model trained with the GroupDRO objective as in Sagawa et al. [85].
- Naive Augmentations - Baseline model on a dataset that also includes augmentations, generated by prompting an LLM to create more examples (without matching or diff-in-diff).
- Conditional Augmentations - Augmentations are generated by matching on auxiliary data and prompting an LLM to create one example in the the style of the other.

The reweighting and MMD approaches are discussed and contrasted to counterfactual augmentation in Section 4. IRM and GroupDRO are the most well-known principled methods for OOD generalization that are used in the literature. The augmentation approaches are compared here to demonstrate the importance of using the causal structure of the data.

## 5.1 Clinical Narratives

**Data.** We consider three representative clinical NLP tasks, *clinical condition* prediction, *note segmentation* and *demographic traits* identification[6], for which we have both ID and OOD data. We utilize several electronic health records (EHR) datasets. We train on MIMIC-III [86], a widely-used medical dataset containing over 2 million notes from $38,597$ adult patients, $49,785$ hospital admissions, and $3,500$ healthcare professionals between 2001 and 2012. MIMIC-III is commonly used in NLP research for clinically-related tasks and for pre-training language models for the medical domain [87]. When available, we use i2b2 2006 and 2010 competitions as our held-out hospital dataset. In the note segmentation task, we use private held-out data.

**Generating notes from counterfactual caregivers.** To generate augmentations, we select caregivers with multiple patients and notes for more than one patient. For each caregiver-patient pair where both their last progress note and discharge summary were written by that caregiver[7], we match them to similar patients having the same initial caregiver but a different one for their discharge summary. In matching, we select patients with similar medications and lab results (denoted as patient's

---

[5]Appendix B includes results where the Baseline model is also BioBERT, SentenceBERT or GPT3.

[6]See Appendix B for results on the *demographic traits* identification task.

[7]During a patient's stay, progress notes capture its current state. When leaving the hospital, a discharge summary is written.

auxiliary data $m$ in Table 1). We then generate counterfactual discharge summaries for matched patients using Algorithm 1(A) and train the model using original data and generated counterfactuals.

Figure 3 presents results for *CATO* (A) using language model representations generated using these matched examples. See Appendix B for training details and results for *CATO* (A) with LLM prompts, and Appendix C for synthetic note examples and the prompts used.

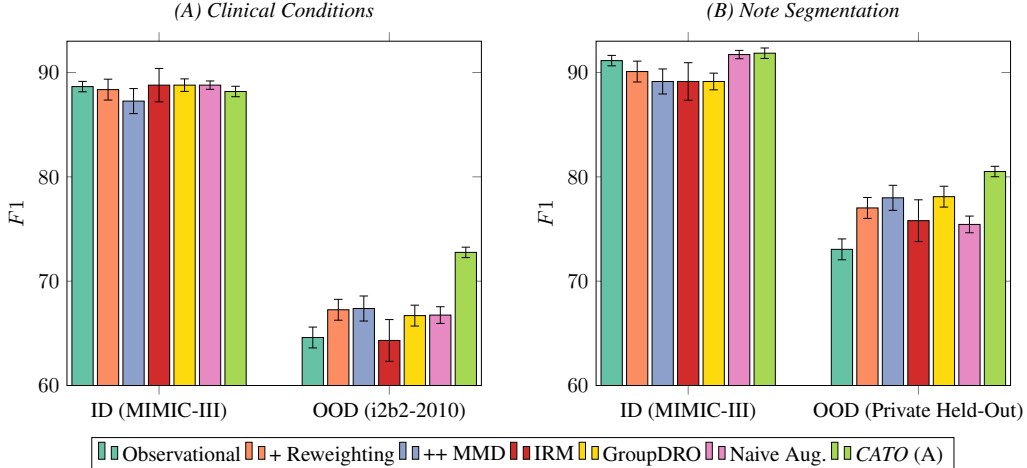

Figure 3: Results ($F1$ averaged across 5 runs) for predicting *clinical conditions* (A) and for clinical *note segmentation* (B) from the text narratives. *CATO* (A) outperforms all baselines on OOD data.

**Clinical Condition Prediction.** *Clinical condition* prediction is a concept extraction task focused on medical concepts in patient reports [88]. Here we trained *PubMED BERT* models on a subset of MIMIC-III, labelled using the same annotation guidelines as in i2b2-2010, the OOD dataset the models are tested on. As can be seen in the Figure 3(A), in the ID setting only the naive augmentations improve performance slightly. In the OOD setting, all OOD methods help (*reweighting*, *MMD*, *IRM*, *GroupDRO*, *CATO* (A)), but our causally-motivated augmentation approach is substantially better than the alternatives. On average (across 5 runs), *CATO* (A) improves precision above the baseline by more than $7\%$ (absolute), and recall by more than $8\%$. The naive augmentation approach improves over the vanilla *PubMED BERT* model, but is outperformed by all OOD methods.

**Note Segmentation.** In this task, models need to recognize sections in free-form clinical notes [89]. Given that section headers vary between hospitals, the models must discern sections based solely on the note content, excluding headers. As can be seen in Figure 3(B), similarly to *clinical condition* prediction, the diff-in-diff approach to augmentations (*CATO* (A)) substantially improved OOD performance, and as expected does not help ID. The naive augmentations are the best performing method ID, but is again outperformed by all other methods OOD.

## 5.2 Restaurant Reviews

**Data.** We use the *CEBaB* dataset [49], which consists of short restaurant reviews and ratings from OpenTable, including evaluations for food, service, noise, ambiance, and an overall rating. We used the train-exclusive split of the dataset, which contains $1,755$ examples. We construct two experimental settings: the original *CeBAB* dataset, and a modified version, denoted as *CeBAB*-Spurious, where there's a spurious correlation between training and deployment.

To construct *CeBAB*-Spurious, we leverage the availability of both the original and perceived ratings for each review in *CeBAB*. The original rating represents the reviewer's initial thoughts when writing the review, while the perceived rating indicates whether the review contains information

| Method | *CeBAB* | *CeBAB*-Spur. |
|---|---|---|
| Observational | **0.85** | 0.64 |
| Reweighting | 0.84 | 0.68 |
| Naive Aug. | 0.80 | 0.62 |
| Conditional Aug. | 0.84 | 0.70 |
| *CATO* (B) | 0.84 | **0.75** |

Table 2: Accuracy on *CeBAB* and *CeBAB*-Spurious. *CATO* (B) outperforms all baselines when we introduce a spurious correlation.

about various restaurant attributes (e.g., food, service, noise, ambiance) and their associated sentiment. We utilize this unique data structure to capture reviewers' writing styles. Some reviewers are concise and provide limited descriptions, while others are more descriptive and include more information. To incorporate this variability, we introduce a new attribute called *food-mention* to signify the presence of food-related information in a review. If the perceived food rating is either negative or positive, we assign a value of $1$ to the *food-mention* attribute; otherwise, it is set to $0$. We subsample the data such that there is a correlation of $0.72$ between *food-mention* and the outcome.

**Generating reviews with counterfactual food mentions.** Following Algorithm 1, we generate counterfactual restaurant reviews conditional on food and overall ratings. We find matched examples for each review, select those with different food-mentions, and prompt an LLM to rewrite them, reflecting how the reviews would appear if the reviewer was more/less concise.

**Results.** As shown in Table 2, adding counterfactual augmentations leads to better OOD generalization, while naive data augmentation hurts model performance In line with the sample complexity argument in Section 4, conditional augmentation effectively doesn't add new data and therefore doesn't improve model performance.

## 5.3 Synthetic Data

To test sensitivity of *CATO* to quality of counterfactuals (Q#4), we generate synthetic data for a binary classification problem where $K = 8$ (cardinality of $C$). We sample $\tilde{P}(C \mid Y)$ to simulate varying degrees of spurious correlations. Then we draw $\mathbf{x} = [\mathbf{x}^*, \mathbf{x}_{\text{spu}}]$ from a Gaussian distribution,

$$\mathbf{x}_i = \begin{bmatrix} \mathbf{x}_i^* \\ \mathbf{x}_{\text{spu},i} \end{bmatrix} \sim \mathcal{N}\left( \begin{bmatrix} \boldsymbol{\mu}_{y_i} \\ \boldsymbol{\mu}_{c_i} \end{bmatrix}, \begin{bmatrix} \sigma^2 \mathbf{I}_{d^*} & 0 \\ 0 & \sigma^2_{\text{spu}} \mathbf{I}_{\mathbf{d_c}} \end{bmatrix} \right).$$

In this case $\hat{\mathbf{x}}_i(c)$ is obtained by adding $\mu_c - \mu_{c_i}$ to $\mathbf{x}_{\text{spu},i}$. To corrupt our augmentation, we instead add $\xi_i (\mu_c - \mu_{c_i})$ where $\xi_i$ is drawn from a truncated Gaussian centered at $\lambda \in (0,1)$. We train models with a fixed sample size (in the appendix we also examine varying sample sizes and additional types of corruption) and evaluate the trained models' accuracy on $P_\perp$ to examine the interplay between spurious correlation strength (measured by mutual information $I(Y;C)$), and counterfactual augmentation quality. As can be seen in Figure 4, corruptions degrade performance under stronger spurious correlations, though a strong corruption is required for reweighting to become preferable.

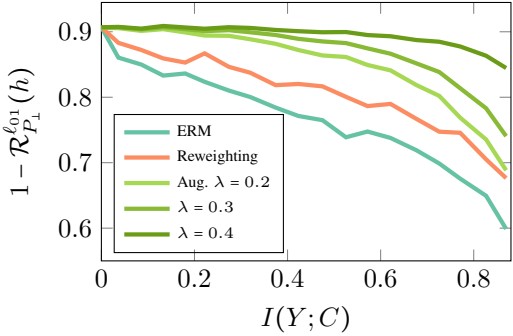

Figure 4: OOD accuracy $(1 - \mathcal{R}^{l_{01}}_{P_\perp}(h))$ and $Y, C$ correlation strength $(I(Y;C))$. *Lower values* of $\lambda$ correspond to *stronger corruptions* of the augmentations. Even with substantial corruption $(\lambda = 0.2)$ and strong correlation, augmentations outperform baselines.

## 6 Discussion

In this work, we have presented a data augmentation approach based on the causal structure of auxiliary data for improving OOD generalization, specifically focusing on text classification tasks. However, our approach is not without limitations. The validity of our assumptions, the specification of the causal graph and the quality of the counterfactual approximation all present challenges to address in future work. Further, our results suggest that performing data augmentation in an unprincipled manner can also hurt model performance. Utilizing additional techniques for OOD generalization, learning the causal structure directly from the data, and improving quality and reliability of the counterfactual approximation process can help mitigate these concerns. Overall, we believe that causally-motivated data augmentation methods like ours can help address challenges in developing robust and reliable machine learning systems, particularly in safety-critical applications.

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
