# Appendix

## A Proofs of Formal Claims

**Notation.** We will use random variables $C, Y, M, X$ with images $[K], \mathcal{Y} = [L], \mathcal{M}, \mathcal{X}$ respectively in our probabilistic causal models. For a function $\tau_c : \mathcal{X} \times \mathcal{M} \to \mathcal{X}$, and measure $P$ over sets in $\mathcal{X} \times \mathcal{M}$, we denote by $\tau_{c,*} P(X, M)$ the pushforward measure [90, §1.4]. $\tau_c(\cdot)$ will be used to refer to the $c$-th coordinate of the output of a function $\tau : \mathcal{X} \times \mathcal{M} \to \mathcal{X}^K$. The notation $\mathcal{H}$ will be used for hypothesis classes where $h : \mathcal{X} \to \mathcal{Y}$ for any $h \in \mathcal{H}$. The $0 - 1$ loss $\ell_{01} : \mathcal{Y} \times \mathcal{Y} \to \{0, 1\}$ is given by $\ell_{01}(\hat{y}, y) = 1_{\hat{y} \neq y}$. For a node $V$ in a causal graph we will use $pa(V)$ for its causal parents.

For completeness we rewrite the definition of our data generating process from the main paper, this time adding the auxiliary data $M$ into our model.

**Definition 1.** *Consider a probabilistic causal model with endogenous random variables $X, X^*, Y, C, M$ taking on values in $\mathcal{X}, \mathcal{X}^*, [L], [K], \mathcal{M}$ and exogenous independent random variables [83] $N_X, N_{X^*}, N_Y, N_C, N_M$, where the induced graph is a DAG that satisfies the following,*

- *$Y$ is $d$-separated from $X$ by $X^*, C, M$ and also by $X^*, C$.*

- *$Y, X^*$ are not descendants of $C$.*

*An anti-causal prediction problem with a spuriously-correlated attribute is a set of distributions $\mathcal{P}$ obtained by all interventions on $C$ that replaces the distribution of exogenous noise $N_C$, mechanism $f_C(pa(C), N_C)$ with another mechanism (i.e. a measurable function $\tilde{f}(pa(C), N_C)$), or sets a fixed value (i.e. $do(C = c)$). Under the settings of this problem, a learner is provided with a set $\{(\mathbf{x}_i, y_i, c_i)\}_{i=1}^N$ sampled i.i.d from $P_{train} \in \mathcal{P}$.*

We denote by $P_\perp \in \mathcal{P}$ the distribution obtained by intervening on $C$ and setting it to a uniform distribution, i.e. $P_\perp(X, X^*, Y, C, M) = K^{-1} \sum_{c \in [K]} P(Y, X, X^*, M \mid do(C = c))$. Note that the problem described by fig. 1 and definition 1 of the main paper is a special case of this setting where $M$ is discarded, and $P_\perp$ coincides with setting $\tilde{P}(C \mid Y)$ to a uniform distribution.

Recall our assumption about perfect recovery of $X^*$.

**Assumption 3.** *For an anti-causal prediction problem with a spuriously correlated attribute, we assume that $X^* = e(X)$ a.e. for some $e : \mathcal{X} \to \mathcal{X}^*$.*

Under these conditions $h(\mathbf{x}) = \arg\max_{y \in [L]} P_\perp(Y = y \mid X = \mathbf{x})$ is an optimal risk-invariant predictor as described below.

**Lemma 1.** *For the prediction problem in definition 1, the Bayes optimal classifier under the unconfounded distribution $P_\perp \in \mathcal{P}$ where $C$ is uniformly distributed and independent of $Y$ is $h^*(\mathbf{x}) = \arg\max_{y \in [K]} P_\perp(Y = y \mid X^* = e(\mathbf{x}))$. It is a minimizer of $\min_{h:\mathcal{X} \to [L]} \max_{P \in \mathcal{P}} \mathcal{R}_P^{\ell_{01}}(h)$ and $\mathcal{R}_P^{\ell_{01}}(h^*) = \mathcal{R}_{P_\perp}^{\ell_{01}}(h^*)$ for all $P \in \mathcal{P}$.*

*Proof.* Assume $P_{\text{train}} \in \mathcal{P}$ is the distribution from which our training data is obtained. We will show that any hypothesis satisfying $h(X) = g \circ e(X)$ for some $g : \mathcal{X}^* \to \mathcal{Y}$ (i.e. that only depends on $X^*$) achieves the same risk over all $P \in \mathcal{P}$. To this end note that for such a hypothesis we have,

$$
\begin{aligned}
R_{P_{\text{train}}}^{\ell_{01}}(h) &= \int \ell_{01}(h(X), Y) P_{\text{train}}(X \mid Y, C, X^*, M) P_{\text{train}}(Y, C, X^*, M) dX^* dX dY dC dM \\
&= \int \ell_{01}(g \circ e(X), Y) P_{\text{train}}(X \mid C, X^*, M) P_{\text{train}}(Y, C, X^*, M) dX^* dX dY dC dM \\
&= \int \ell_{01}(g(X^*), Y) P_{\text{train}}(X \mid C, X^*, M) P_{\text{train}}(Y, C, X^*, M) dX^* dX dY dC dM \\
&= \int \ell_{01}(g(X^*), Y) P_{\text{train}}(X^*, Y) dX^* dY \\
&= \int \ell_{01}(g(X^*), Y) P(X^*, Y) dX^* dY.
\end{aligned}
$$

The first line writes down the expected risk explicitly, the second removes conditioning on $Y$ in the distribution on $X$ since we assumed $Y$ is $d$-separated from $X$ by $C, X^*, M$. In the third line we

make it explicit that $h$ depends on $X^*$ alone, then we integrate out $X, C, M$. On the last line we remove the subscript train to denote that this distribution in fixed across $P \in \mathcal{P}$ as we assumed that $X^*, Y$ are non-descendants of $C$ (and members of $\mathcal{P}$ are obtained by interventions on $C$). Now for any $\tilde{P} \in \mathcal{P}$ we may repeat this derivation for $R_{\tilde{P}}^{l_{01}}(h)$ and we will obtain the same term (since $P(X^*, Y)$ are fixed regardless of the intervention applied in $P$, as we just argued), and we may conclude $R_{P_{\text{train}}}^{\ell_{01}}(h) = R_{\tilde{P}}^{\ell_{01}}(h)$.

Next to show that the Bayes optimal classifier over $P_\perp$ is the min-max optimal classifier w.r.t $\mathcal{P}$, consider the interventional distribution where $C$ is set to some fixed value $c \in [K]$, i.e. $P(X, X^*, Y \mid do(C = c))$. Under the graph we obtain from this intervention, $Y$ is $d$-separated from $X$ given $X^*$. Hence,

$$P(Y \mid X = \mathbf{x}, do(C = c)) = \int_{X^*} P(Y \mid X^*, X = \mathbf{x}, do(C = c)) P(X^* \mid X = \mathbf{x}, do(C = c)) dX^*$$
$$= P(Y \mid X^* = e(\mathbf{x}), X = \mathbf{x}, do(C = c))$$
$$= P(Y \mid X^* = e(\mathbf{x}), do(C = c)),$$

where the first equality holds since $X^* = e(X)$ and the second from $d$-separation. Hence the Bayes optimal classifier under $P(Y, X \mid do(C = c))$ is $h^*(\mathbf{x}) = g \circ e(\mathbf{x}) = \arg\max_{y \in [L]} P(Y = y \mid e(\mathbf{x}), do(C = c))$. As argued earlier, since $Y, X^*$ are non-descendants of $C$, it holds that $P(Y \mid e(X), do(C = c))$ is fixed across all $c \in [K]$. Hence $h^*(\mathbf{x})$ is the Bayes optimal classifier for all such interventional distributions and also for $P_\perp(X, Y) = \frac{1}{K} \sum_{c \in [K]} P(X, Y \mid do(C = c))$, and from our earlier discussion it is risk-invariant, i.e. $R_{P_\perp}^{\ell_{01}}(h^*) = R_P^{\ell_{01}}(h^*)$ for all $P \in \mathcal{P}$, which also means $\max_{p \in \mathcal{P}} R_P^{\ell_{01}}(h^*) = R_{P_\perp}^{\ell_{01}}(h^*)$. It is the min-max optimal classifier w.r.t $\mathcal{P}$ since any $h \neq h^*$ will have $\max_{p \in \mathcal{P}} R_P^{\ell_{01}}(h) \geq R_{P_\perp}^{\ell_{01}}(h) \geq R_{P_\perp}^{\ell_{01}}(h^*)$. $\square$

Next we turn to prove a bound on sample complexity of counterfactual data augmentations.

**Lemma 2.** *Consider an anti-causal prediction problem with a spuriously-correlated attribute (definition 1), a measurable function $\tau : \mathcal{X} \times \mathcal{M} \to \mathcal{X}^K$, and let $d_1(P, Q)$ denote the total variation distance between two distributions $P, Q$. Further let $h^*$ denote the optimal hypothesis w.r.t $\mathcal{R}_{P_\perp}^{\ell_{01}}$ and let $\lambda_{aug} = \left[ R_{aug}^{\ell_{01}}(h^*) + R_{P_\perp}^{\ell_{01}}(h^*) \right]$. For any hypothesis $h \in \mathcal{H}$, and any $\delta \in (0.5, 1)$ it holds that with probability at least $1 - \delta$ over the draw of the training set,*

$$\mathcal{R}_{P_\perp}^{\ell_{01}}(h) \leq \widehat{\mathcal{R}}_{aug}^{\ell_{01}}(h) + \sqrt{\frac{\log(1/\delta)}{N}} + K^{-1} \cdot \sum_{c \in [K]} d_1\left(\tau_{c,*}\left(P_{train}(X, M)\right), P\left(X(c)\right)\right) + \lambda_{aug}.$$

*Proof.* Our first step is to show that for any hypothesis $h \in \mathcal{H}$, if our augmentation process is exact in the sense that $\tau_c(X, M) = X(c)$ a.e., then the expected risk (i.e. risk taken over an infinitely large sample) on the augmented data coincides with that over the unconfounded distribution $P_\perp(X, Y) = P_{\text{unif}}(C)P(X, Y \mid do(C))$.

$$\mathcal{R}_{\text{aug}}^{\ell_{01}}(h) = \mathbb{E}_{P_{\text{train}}(C, Y, M, X)}\left[ K^{-1} \sum_{c \in [K]} \ell_{01}(h\left(\tau_c(X, M)\right), Y) \right]$$
$$= K^{-1} \sum_{c \in [K]} \mathbb{E}_{P_{\text{train}}(C, Y, M, X)}[\ell_{01}(h\left(X(c)\right), Y)]$$
$$= K^{-1} \sum_{c \in [K]} \mathbb{E}_{P_{\text{train}}(C, Y, X)}[\ell_{01}(h\left(X(c)\right), Y(c))]$$
$$= K^{-1} \sum_{c \in [K]} \mathbb{E}_{P(Y, X \mid do(C = c))}[\ell_{01}(h\left(X\right), Y)]$$
$$= \mathcal{R}_{P_\perp}^{\ell_{01}}(h). \tag{3}$$

To bound $\mathcal{R}_{\text{aug}}^{\ell_{01}}(h) - \hat{\mathcal{R}}_{\text{aug}}^{\ell_{01}}(h)$ we note that $\{\mathbf{x}_i, y_i, \mathbf{m}_i\}_{i=1}^N$ are $i.i.d$ samples from a joint distribution, where we may consider the loss on each example as $K^{-1} \sum_{c \in [K]} \ell_{01}(h(\tau_c(\mathbf{x}_i, \mathbf{m}_i), y_i))$, then by

standard results using the Hoeffding inequality, e.g. Mohri et al. [91, Corollary 2.11], we get that for $\delta \in (0.5, 1)$,

$$\mathcal{R}_{\text{aug}}^{\ell_{01}}(h) \le \widehat{\mathcal{R}}_{\text{aug}}^{\ell_{01}}(h) + \sqrt{\frac{\log(1/\delta)}{N}}. \tag{4}$$

Finally, to obtain our result consider any $c \in [C]$. Denote

$$\mathcal{R}_{\text{aug},c}^{\ell_{01}}(h) := \mathbb{E}_{P_{\text{train}}(Y,M,X)}[\ell_{01}(h(\tau_c(X,M))Y)],$$
$$\mathcal{R}_{P_\perp,c}^{\ell_{01}}(h) := \mathbb{E}_{P(Y,X|do(C=c))}[\ell_{01}(h(X),Y)],$$

and for $h^*$ denote $\mathcal{R}_{\text{aug},c}^{\ell_{01}}(h,h^*) := \mathbb{E}_{P_{\text{train}}(Y,M,X)}[\ell_{01}(h(\tau_c(X,M)), h^*(\tau_c(X,M)))]$ and respectively for $\mathcal{R}_{P_\perp}^{\ell_{01}}(h,h^*)$. The rest of our derivation is along the lines of Ben-David et al. [82, Theorem 2]. We use the distance

$$d_{\mathcal{H}\Delta\mathcal{H}}(\tau_{c,*}P_{\text{train}}(X,M), P(X(c))) = 2 \sup_{g \in \mathcal{H}\Delta\mathcal{H}} |P_{\text{train}}(g(\tau_c(X,M)) = 1) - P(g(X(c)) = 1)|,$$

where $\mathcal{H}\Delta\mathcal{H} = \{g(\mathbf{x}) = 1_{h(\mathbf{x}) \ne h'(\mathbf{x})} \mid h, h' \in \mathcal{H}\}$ is a set of binary hypotheses, i.e. functions that mark disagreements between hypotheses in $\mathcal{H}$. It is easy to see that $d_{\mathcal{H}\Delta\mathcal{H}}$ lower bounds $d_1$ which takes the supremum w.r.t all measurable subsets for the two measures, since the sets of inputs where $h(\mathbf{x}) = 1$ are contained in those subsets. Also from [82, Lemma 3] we have that for any hypotheses $h, h' \in \mathcal{H}$ it holds that

$$\left| R_{\text{aug},c}^{l_{01}}(h,h') - R_{P_\perp,c}^{l_{01}}(h,h') \right| \le \frac{1}{2} d_{\mathcal{H}\Delta\mathcal{H}}(\tau_{c,*}P_{\text{train}}(X,M), P(X(c)))$$

Then following the proof in Ben-David et al. [82, Theorem 2], where the first and third inequalities will rely on the triangle inequality for classification errors [92], we may get:

$$\begin{aligned}
\mathcal{R}_{P_\perp,c}^{\ell_{01}}(h) &\le \mathcal{R}_{P_\perp,c}^{\ell_{01}}(h^*) + \mathcal{R}_{P_\perp,c}^{\ell_{01}}(h,h^*) \\
&\le \mathcal{R}_{P_\perp,c}^{\ell_{01}}(h^*) + \mathcal{R}_{\text{aug},c}^{\ell_{01}}(h,h^*) + [\mathcal{R}_{P_\perp,c}^{\ell_{01}}(h,h^*) - \mathcal{R}_{\text{aug},c}^{\ell_{01}}(h,h^*)] \\
&\le \mathcal{R}_{P_\perp,c}^{\ell_{01}}(h^*) + \mathcal{R}_{\text{aug},c}^{\ell_{01}}(h,h^*) + \frac{1}{2} d_{\mathcal{H}\Delta\mathcal{H}}(\tau_{c,*}P_{\text{train}}(X,M), P(X(c))) \\
&\le \mathcal{R}_{\text{aug},c}^{\ell_{01}}(h) + \mathcal{R}_{P_\perp,c}^{\ell_{01}}(h^*) + \mathcal{R}_{\text{aug},c}^{\ell_{01}}(h^*) + \frac{1}{2} d_{\mathcal{H}\Delta\mathcal{H}}(\tau_{c,*}P_{\text{train}}(X,M), P(X(c))) \\
&= \mathcal{R}_{\text{aug},c}^{\ell_{01}}(h) + \mathcal{R}_{P_\perp,c}^{\ell_{01}}(h^*) + \mathcal{R}_{\text{aug},c}^{\ell_{01}}(h^*) + \frac{1}{2} d_{\mathcal{H}\Delta\mathcal{H}}(\tau_{c,*}P_{\text{train}}(X,M), P(X(c)))
\end{aligned}$$

Finally, we note that $\mathcal{R}_{P_\perp}^{\ell_{01}}(h) = K^{-1} \sum_{c \in [K]} \mathcal{R}_{P_\perp,c}^{\ell_{01}}(h)$ and similarly we have that $\mathcal{R}_{\text{aug}}^{\ell_{01}}(h) = K^{-1} \sum_{c \in [K]} \mathcal{R}_{\text{aug},c}^{\ell_{01}}(h)$, hence applying the above inequality for all $c \in [K]$ and averaging we get:

$$\begin{aligned}
\mathcal{R}_{P_\perp}^{\ell_{01}}(h) &\le \mathcal{R}_{\text{aug}}^{\ell_{01}}(h) + \frac{1}{2} K^{-1} \sum_{c \in [K]} d_{\mathcal{H}\Delta\mathcal{H}}(\tau_{c,*}P_{\text{train}}(X,M), P(X(c))) + \lambda_{\text{aug}} \\
&\le \mathcal{R}_{\text{aug}}^{\ell_{01}}(h) + K^{-1} \sum_{c \in [K]} d_1(\tau_{c,*}P_{\text{train}}(X,M), P(X(c))) + \lambda_{\text{aug}}.
\end{aligned}$$

Combining with eq. (4) we get the desired result. $\qquad \square$

## A.1 Additional Causal Structures Where our Approach may be Used

The problem setting we analyze in this work (see definition 1) captures a few interesting problems, mainly described as shortcut learning in the literature [25, 93, 94]. However counterfactual data augmentation, and subsequently our approach of using auxiliary data to perform it, are applicable to additional problem settings. Wang and Veitch [72] formalize domain-invariant learning under many data generating processes they refer to as Causally Invariant with Spurious Associations (CISA), where $Z$ (in our setting the caregiver $C$) is called the spurious factor of variation. These settings include a variety of causal and anti-causal prediction problems, and they assume that there exists some part of the input $X$, referred to as $X_Z^\perp$, that holds all the information in $X$ that is not caused by $Z$. Whenever it holds that $Y \perp\!\!\!\perp X \mid X_Z^\perp, Z$ the association between $Z$ and $Y$ is called "purely

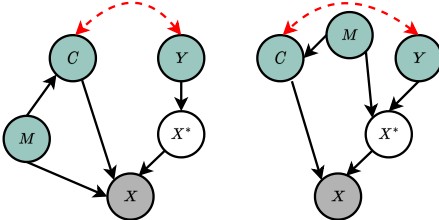

Figure 5: Possible causal structures that involve the auxiliary data $M$, where unobserved $M$ corresponds to unobserved confounding between $X$ and $C$.

spurious" and Thm. 9 in Wang and Veitch [72] states that for all such problems counterfactual data augmentation learns the optimal invariant predictor over the training distribution. Hence in all such settings, improving counterfactual data augmentation with *CATO* can be beneficial towards OOD generalization. We refer the interested reader to [72] for further details on CISA problems and their properties.

We further note that in our work we excluded the auxiliary data $M$ from the causal model as we are agnostic to its specific causal relation with other factors in the data, so long as it satisfies **??** 1 of strong ignorability. fig. 5 depicts two potential structures that may adhere to this assumption.

## B   Experimental Details

We provide here further details about the experimental setup, the datasets we use, hyperparameters chosen for training the models, and data splits. We also include additional experiments that were omitted from the main paper for brevity, including experiments on identifying *demographic traits* in clinical narratives.

### B.1   Clinical Narratives

#### B.1.1   Data

We describe here the *MIMIC-III i2b2-2006* and *i2b2-2010* datasets.

**MIMIC-III.**   The *MIMIC-III* (Medical Information Mart for Intensive Care III) dataset is a large, publicly available database containing detailed and anonymized health-related data associated with over 40,000 patients who stayed in critical care units at the Beth Israel Deaconess Medical Center in Boston, Massachusetts between 2001 and 2012. *MIMIC-III* is a rich resource for researchers in various fields, such as medicine, data science, artificial intelligence, and healthcare analytics. The dataset contains a diverse range of data types, including demographics, vital signs, laboratory test results, medications, and clinical notes. The dataset contains over 2 million clinical notes contributed by over $3,500$ distinct healthcare professionals, including doctors, nurses, and other clinicians, with an average of 571 notes per author.

The notes in the *MIMIC-III* dataset come in various types, reflecting the diverse aspects of patient care and documentation in the intensive care setting. Some of the most common note types include:

- Nursing/Progress notes: These are daily notes written by nurses or other care providers, documenting the patient's progress, condition, and care provided.

- Radiology reports: Reports written by radiologists after interpreting medical imaging studies (e.g., X-rays, MRIs, CT scans).

- ECG reports: Reports documenting the interpretation of electrocardiogram results.

- Discharge summaries: Comprehensive summaries written by physicians when a patient is discharged from the hospital, outlining the patient's hospital course, treatments, and follow-up instructions.

- Physician consult notes: Notes written by specialists when consulted by the primary care team to provide their expert opinion on specific medical issues.

- Pharmacy notes: Notes documenting medication-related information, including dosing, administration, and potential drug interactions.

- Social work notes: Notes related to the patient's psychosocial status, including social and family support, living arrangements, and other relevant factors.

**i2b2-2006.** The i2b2 (Informatics for Integrating Biology and the Bedside) initiative is a collaborative effort that aims to develop new methods and tools for biomedical research. It focuses on the development of a scalable computational infrastructure that can be used to accelerate the translation of basic research findings into clinical applications. As part of this effort, i2b2 has hosted several shared tasks and challenges related to natural language processing and machine learning in healthcare.

In 2006, the first i2b2 challenge, known as the *i2b2-2006* challenge, was conducted, focusing on the identification of obesity and its comorbidities in discharge summaries. The dataset provided for the challenge contained $694$ de-identified discharge summaries, which were randomly selected from the Research Patient Data Registry (RPDR) at Partners HealthCare. The dataset was divided into a training set of $514$ discharge summaries and a test set of $180$ discharge summaries. It is important to mention that the *i2b2-2006* dataset is relatively small compared to the *MIMIC-III* dataset and does not provide detailed information about the number of distinct authors or the average number of notes per author.

However, the discharge summaries typically include various sections such as patient demographics, admission and discharge dates, admission diagnoses, hospital course, procedures, medications, and follow-up plans. These summaries are generally written by physicians at the time of patient discharge, providing an overview of the patient's medical condition, treatment received, and overall hospital stay.

**i2b2-2010.** The *i2b2-2010* challenge, also known as the i2b2/VA challenge, was a shared task organized by the i2b2 (Informatics for Integrating Biology and the Bedside) initiative in collaboration with the US Department of Veterans Affairs (VA). The challenge aimed to encourage the development of natural language processing (NLP) and machine learning techniques for extracting medical concepts from clinical narratives. Specifically, the *i2b2-2010* challenge focused on the identification of medical problems, tests, and treatments from free-text clinical records.

The dataset provided for the *i2b2-2010* challenge contained $826$ de-identified clinical records, which were sourced from three different institutions: Partners HealthCare, the University of Pittsburgh Medical Center (UPMC), and the VA. The dataset was divided into a training set of $349$ records and a test set of $477$ records.

Similar to the *i2b2-2006* challenge, the *i2b2-2010* dataset is relatively small compared to the *MIMIC-III* dataset and does not provide detailed information about the number of distinct authors or the average number of notes per author. The clinical records in the dataset are composed of diverse note types, such as discharge summaries, progress notes, radiology reports, and pathology reports, contributed by physicians, nurses, and other healthcare professionals.

While the dataset does not provide specific information about the number of distinct authors, the fact that the notes were contributed by different types of healthcare professionals across multiple institutions increases the dataset's diversity, making it more representative of real-world clinical settings.

### B.1.2   PubMED BERT

In our clinical narratives experiments, we use *PubMED BERT* [84], a variant of of the original BERT model [95], as our vanilla model. That is, all of the baselines and *CATO* all use it either for embedding clinical text or for predicting *conditions*, *demographic traits* and *note segments*.

*PubMED BERT* is a BERT-based (Bidirectional Encoder Representations from Transformers) model that has been pre-trained specifically on biomedical and scientific text data [84]. The model leverages the BERT architecture, which is a transformer-based deep learning model that has gained significant attention in natural language processing (NLP) for its state-of-the-art performance across a wide range of tasks.

*PubMED BERT* is pre-trained on a large corpus of approximately 14 million biomedical abstracts from the PubMed database, which is a comprehensive repository of biomedical literature. By pre-training the model on domain-specific data, *PubMED BERT* is expected to have a better understanding of biomedical concepts, terminology, and language patterns compared to general domain models like BERT-base and BERT-large [95].

The main advantage of using *PubMED BERT* for biomedical text mining tasks is its domain-specific knowledge, which can lead to improved performance and more accurate results when fine-tuned on various downstream tasks, such as named entity recognition, relation extraction, document classification, and question answering. Since *PubMED BERT* is pre-trained on a large corpus of biomedical text, it is better suited to capturing the unique language patterns, complex terminology, and the relationships between entities in the biomedical domain.

**Hyperparameters for Fine-Tuning PubMED BERT on *MIMIC-III*.** In our study, we leveraged a pre-trained *PubMED BERT* model and fine-tuned it on the *MIMIC-III* dataset. During pre-training, the model employed masked language modeling and next sentence prediction objectives. The architecture consisted of 12 layers, 768 hidden units, and 12 attention heads. For task-specific optimization, we used the following hyperparameters: a $3e-5$ learning rate with a linear warmup during the initial 10% of training steps, a batch size of 32, a maximum sequence length of 512 tokens, and a dropout rate of 0.1. The AdamW optimizer was applied with a 0.01 weight decay and a 1.0 gradient clipping threshold. To prevent overfitting, early stopping was based on validation loss and used a 3-epoch patience. The fine-tuning process ran for up to 20 epochs, unless early stopping criteria were met sooner.

The fine-tuning process was executed on a high-performance computing cluster with multiple NVIDIA Tesla V100 GPUs, each equipped with 32 GB of memory, using the *PyTorch* deep learning framework [96]. The dataset was preprocessed and tokenized using the *HuggingFace Transformers* library [97].

### B.1.3 *Demographic Traits* Detection

*Demographic Traits* detection is the task of identifying residual private information in the clinical note, after removing the known identifier types (names, ages, dates, addresses, ID's, etc.) [71]. We train all models on a subset of *MIMIC-III* and test on *i2b2-2006*. Table 3 presents our results. While performance gains from the Causal Augmentation approach are not as large as in the other clinical NLP tasks, its is still the best method in terms of $F1$ score on out-of-distribution examples.

|  | ID (*MIMIC-III*) | | | OOD (*i2b2-2006*) | | |
|---|---|---|---|---|---|---|
|  | P | R | F1 | P | R | F1 |
| *PubMED BERT* | 80.61 | 78.12 | 79.34 | 53.32 | 90.1 | 66.92 |
| *+ Re-Weighting* | 81.31 | 78.57 | **79.92** | 56.75 | 91.38 | 70.02 |
| *++ MMD* | 80.68 | 78.84 | 79.75 | 56.19 | **91.49** | 69.62 |
| *Bio BERT* | 79.5 | 77.63 | 78.55 | 53.32 | 89.84 | 66.71 |
| *Sentence BERT* | 79.29 | 76.18 | 76.53 | 52.22 | 89.82 | 65.04 |
| *GPT3* | 78.31 | 76.01 | 77.18 | 52.73 | 88.52 | 63.98 |
| *Naive Aug.* | **81.45** | **79.35** | 80.39 | 52.9 | 89.58 | 66.52 |
| ***Causal Aug.*** | 80.65 | 78.84 | 79.73 | **59.76** | 90.16 | **71.88** |

Table 3: Results (averaged across 5 runs) for predicting demographic traits from the text narratives on in-distribution and out-of-distribution data.

### B.2 Restaurant Reviews

**Data.** We use the *CEBaB* dataset [49], which consists of short restaurant reviews and ratings from OpenTable, including evaluations for food, service, noise, ambiance, and an overall rating. For our experiments, we used the train-exclusive split of the dataset, which contains 1,755 examples.

To analyze the data, we transformed the overall rating into a binary outcome. The original rating scale ranges from 1 to 5, and we classified a rating of 3 or higher as 1, and anything below as 0. We

utilized a bag-of-words model with *CountVectorizer* and fitted logistic regression models from the *sklearn* library [98].

To investigate these questions, we construct two experimental settings: the original *CeBAB* dataset, and a modified version, denoted as *CeBAB*-Spurious, where there's a spurious correlation between training and deployment.

The data is randomly split into a training set with $1,000$ examples and a test set with $755$ examples. We explore two data augmentation schemes:

1. Naive data augmentation: This approach involves randomly selecting two reviews from the dataset and prompting *GPT-4* [99] to rewrite one restaurant review in the style of the other. By applying the naive augmentation, we obtain an additional $1,000$ training examples.

2. Conditional data augmentation : We match the ratings and sub-ratings in the reviews to create pairs. We then prompt *GPT-4* to rewrite one review to match the style of the other. Because not all pairs have matches in this case, the conditional data augmentation generates 926 augmentations. See Appendix B for details of the prompt.

**Generating reviews with counterfactual food mentions.** Following the counterfactual generation procedure in Algorithm 1, we generate counterfactual restaurant reviews conditional on food rating and overall rating. For each review, we first find a set of matched examples. We then select the subset that has different food-mention attribute and prompt *GPT-4* to rewrite. This results in $2,537$ augmentations. The counterfactual augmentation should capture what the reviews should look like had a reviewer been more/less concise. Following Algorithm 1, we generate counterfactual restaurant reviews conditional on food and overall ratings. We find matched examples for each review, select those with different food-mentions, and prompt a *GPT-4* to rewrite them, reflecting how the reviews would appear if the reviewer was more/less concise.

**Prompt Example.**

```
helper_prompt = """
you are a very helpful, diligent, and intelligent language model assistant,
your task to generate counterfactual restaurant reviews,
that is what the restaurant review would be if it is given a different rating.
You will be given an original restaurant review and a comparator review
Your task is to rewrite the original review, such that it will have the same
review score as the comparator review.
The rating is with respect to ambiance, food, noise, and service.
---- EXAMPLE INPUT - START -----

original_review: [],
original_ratings: [
rating_ambiance: score,
rating_food: score,
rating_noise: score,
rating_service: score
]

compare_reviews:[]
compare_ratings:[
rating_ambiance: score,
rating_food: score,
rating_noise: score,
rating_service: score
]

---- EXAMPLE INPUT - END -----
ANSWER FORMAT:
{
```

```
original_review: [],
original_score: [],
rewrite_review: [],
}

"""
```

## B.3 Synthetic Data

As described in the main paper we study a binary classification problem where $K = 8$ (cardinality of $C$), and sample $\tilde{P}(C \mid Y)$ to simulate varying degrees of the spurious correlation (specifically, we draw ). Then we draw $\mathbf{x} = [\mathbf{x}^*, \mathbf{x}_{\text{spu}}]$ from a Gaussian distribution,

$$\mathbf{x}_i = \begin{bmatrix} \mathbf{x}_i^* \\ \mathbf{x}_{\text{spu},i} \end{bmatrix} \sim \mathcal{N}\left( \begin{bmatrix} \boldsymbol{\mu}_{y_i} \\ \boldsymbol{\mu}_{c_i} \end{bmatrix}, \begin{bmatrix} \sigma^2 \mathbf{I}_{d^*} & 0 \\ 0 & \sigma_{\text{spu}}^2 \mathbf{I}_{\mathbf{d_c}} \end{bmatrix} \right).$$

In our simulations, we set $d^* = 10, d_{\text{spu}} = 300$ and $\sigma_{spu}^2 = 0.05, \sigma = 0.01 d^*$ to make the max-margin classifiers depend on the spurious features. The parameters $\mu_{y_i}, \mu_{c_i}$ are drawn uniformly from a sphere of norm $1/3$ and $60$, respectively. For the corruptions of augmentations where we add $\xi_i(\mu_c - \mu_{c_i})$, the $\xi_i$ variables are drawn from a truncated Gaussian centered at $\lambda$ with standard deviation $0.1$.

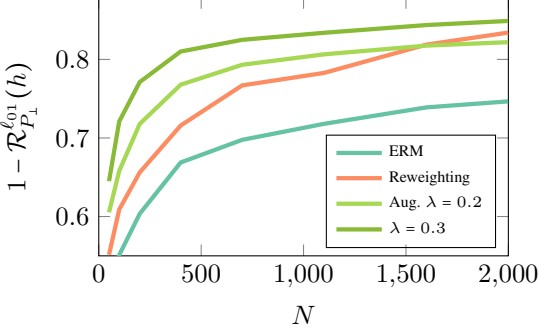

Figure 6: OOD accuracy $(1 - \mathcal{R}_{P_\perp}^{l_{01}}(h))$ for growing size of i.i.d training set $N$. We run 15 repetitions where $\tilde{P}(C \mid Y)$ are drawn randomly with correlation strength $I(Y; C) = 0.743 \pm 0.019$. With large amounts of data, the reweighting method approaches optimal performance and may outperform solutions based on corrupted data augmentation (e.g. it surpasses the more heavily corrupted data augmentation with $\lambda = 0.2$).

For the results in fig. 4 of the main paper we set the number of training examples $N$ at $600$ and the distributions $\tilde{P}(C \mid Y)$ are sampled such that for each interval of size $0.05$ between $0$ and $0.9$ for the values of $I(Y; C)$, we draw 30 instances within that interval. In fig. 6 we give results for another experiment where we plot curves for reweighting, ERM and corrupted augmentation under several values of $N$ under a strong spurious correlation. We draw values for $\tilde{P}(C \mid Y)$ such that that $I(Y; C)$ is in $[0.7, 0.8]$ (mean $0.743$ and standard deviation $0.019$ with 15 repetitions). Considering the bounds in eq. (2) and the one in lemma 2, we expect that as $N$ grows the reweighting method will approach optimal accuracy, while the augmentation methods will incur some bias due to the imperfect approximation of counterfactuals. In fig. 6 we see that the augmentation based methods begin to plateau, while the reweighting method gains larger improvement from the growing $N$, and outperforms the more heavily corrupted augmentation method ($\lambda = 0.2$) with a large enough training set. The best possible mean accuracy in this simulation is $0.934$ (as we gather from training a classifier from $\mathbf{x}^*$ alone), the best accuracy obtained by the methods in the figure is $0.849$ (Aug. $\lambda = 0.3$ with $N = 1999$).

## C   Synthetic Note Examples

While it is not possible to share real notes from the *MIMIC-III* or any of the *i2b2* datasets, we can share how a typical note might look like by generating synthetic notes with LLMs. Here, we generate a synthetic note using *GPT-4,* and then generate a counterfactual note using our prompt:

---

**Discharge Summary Note**
**Dr. Jane Doe**
**March 29, 2023**
**Patient Name:** John Doe
**Admission Date:** March 25, 2023
**Discharge Date:** March 29, 2023
**Diagnosis:** Pneumonia
**History of Present Illness:** Mr. Doe is a 58-year-old male with a past medical history of hypertension who presented to the emergency department with a 3-day history of fever, cough, and shortness of breath. Chest X-ray showed bilateral infiltrates, and he was admitted for treatment of community-acquired pneumonia.
**Hospital Course:** The patient was started on intravenous antibiotics and supplemental oxygen. Over the course of his hospitalization, his oxygen requirements decreased, and his fever and cough resolved. He was able to tolerate a regular diet and was mobilized with physical therapy.
**Consultations:** Infectious disease was consulted, and the patient's antibiotics were adjusted based on culture results.
**Discharge Medications:**

1. Levofloxacin 750mg daily for 10 days

2. Acetaminophen 650mg every 6 hours as needed for fever or pain

3. Albuterol inhaler 2 puffs every 4 hours as needed for shortness of breath

**Follow-up:** The patient was instructed to follow up with his primary care physician in 1 week for a repeat chest X-ray and evaluation of his symptoms.
**Discharge Instructions:**

1. Finish the entire course of antibiotics as prescribed.

2. Use the albuterol inhaler as needed for shortness of breath.

3. Rest and avoid strenuous activity until cleared by your doctor.

4. Follow up with your primary care physician in 1 week.

**Discharge Condition:** The patient was discharged in stable condition with improved symptoms.

---

Here's a counterfactual discharge summary note written by a doctor with a different writing style:

---

**Patient Name:** John Doe
**Admission Date:** March 25, 2023
**Discharge Date:** March 29, 2023
**Diagnosis:** Pneumonia
**HPI:** John is a middle-aged man who came in with a fever and cough that he's had for three days. He was diagnosed with pneumonia and admitted.
**Hospital Course:** John was given antibiotics and oxygen. He slowly got better and was able to eat normally and move around more. He saw an infectious disease specialist who adjusted his treatment.
**Consultations:** Infectious disease saw John and changed his medicine.
**Discharge Medications:**

1. Levofloxacin 750mg once a day for 10 days

2. Acetaminophen 650mg every 6 hours as needed for fever or pain

3. Albuterol inhaler 2 puffs every 4 hours as needed for shortness of breath

**Follow-up:** Follow up with PCP in 1 week.
**Discharge Instructions:**

1. Finish your antibiotics.

2. Use the inhaler if you need it.

3. Rest and avoid heavy activity until you feel better.

4. Follow up with your doctor next week.

**Discharge Condition:** Stable, going home.

---

As can be seen from these examples, the counterfactual note is much more concise and to-the-point than the original example. The language used is more direct and less descriptive, and there is less detail provided about the patient's course of treatment.

## D    Possible Limitations of LLMs in Generating Augmented Datasets

As mentioned in our discussion, there are several possible limitations that should be carefully considered before applying our approach in practice, especially in high-stakes applications such has medical notes classification. We list some of the main possible limitations and points to consider, along with a short discussion on each.

- *LLM generation quality*: LLMs vary in their ability to generate realistic text. It is possible that LLMs introduce biases into our problem, inherited from their own training data. This requires further study, however from our manual examination we found their quality satisfactory (see appendix C for generation examples) and that OOD generalization also improved for models trained on the augmented data they generate. We also include experiments with several types of LLMs in appendix B to verify that our findings are consistent across the types of LLMs we considered.

- *Counterfactual approximation*: Other than generation quality, the additional challenge in using LLMs for counterfactual data augmentation is our ability to elicit a good approximation to the counterfactual text. Our methods rely on principles from causal inference to advance disciplined approaches for this task. While further studies are required (e.g. systematically comparing small sets of manual re-writes of texts to the elicited LLM output), we view our work as a promising first step in this direction, which we expect to be significantly extended and improved in future work.

- *Effect of biases on OOD generalization*: Since we focus on OOD generalization, the limitations and possible biases mentioned above must be weighed within this context. Namely, we should bear in mind that even though generation may be biased, this bias is only harmful when it affects the generalization of a downstream classifier, and this is what we evaluate. Further, in OOD generalization we consider cases where the training data is biased in the first place, and training a standard predictive model also results in a biased solution. Hence we must weigh risks and limitations of alternative solutions vs. those of LLMs.