# OpenReview forum: "Data Augmentations for Improved (Large) Language Model Generalization"
_NeurIPS.cc/2023/Conference — NeurIPS 2023 poster_

### Official Review · Reviewer_8nR3 · 2023-07-06

**Soundness:** 2 fair
**Presentation:** 3 good
**Contribution:** 3 good
**Rating:** 5
**Confidence:** 3

**Summary:**

This paper focuses on counterfactual data augmentation for training robust machine learning models on text data. The concerned applications such as healthcare are safety-critical, highlighting the importance of this work. To generate data, the authors propose to employ a Large Language Model (LLM) to rewrite the documents. The authors also propose utilizing auxiliary data in the generation process to encourage robustness. In the experiments, the proposed method is proved to be both robust and effective.

**Strengths:**

1. The research topic, training robust classifiers on safety-critical applications, is clearly an important task. Therefore, the positive experiment results as reported are expected to bring practical value to the real world.
2. The model performance reported in the experiment section is good. Normally an out-of-distribution generalizable model sacrifices performance in the in-distribution setting. It seems that the proposed method does not suffer from this issue.
3. The paper is overall well-written and easy to follow. The experiment settings in the main paper and Appendix are detailed.

**Weaknesses:**

1. The novelty of the proposed method may be limited. It seems that the main difference between the existing and the proposed method is introducing auxiliary data, which may not be considered as a significant technical advance.
2. The competitors in the experiments are rather basic. In particular, it may be worth considering including IRM (Arjovsky et al., 2019) and GroupDRO (Sagawa et al., 2019) (or Just-Train-Twice (Liu et al., 2021) if the authors are interested in re-weighting) for comparison.
3. As this research targets at safety-critical applications, there are some details regarding model bias and robustness may require more clarification. The questions are in "question to authors".

**Questions:**

1. The data augmentation process relies on an off-the-shelf LLM. Is it possible that the LLM can bring additional and uncontrollable biases to the proposed method?
2. Following the first question, how critical is the selection of LLM in this work?
3. How critical is the quality of the auxiliary data? Also, would it be difficult to identify informative auxiliary data in practice?
4. Regarding the writing,
- line 134: Should the term under argmax be c \in [K] instead of y \in [K]?
- Would it be more clear if the authors specify that the employed LLM is GPT4 in the main text instead of Appendix?

**Limitations:**

 The authors adequately address the limitation.

---

> ### Author Rebuttal · Authors · 2023-08-10
>
> Thank you for reviewing our paper and recognizing positive points on the importance of the topic we study, the achieved empirical performance and writing quality. In what follows we aim to address each question and concern in your review.
>
> **Limited novelty**:
> The novelty of our work, besides the favorable performance (w.r.t to all baselines) on a large scale real-world task, is:
> 1. Introducing causal methods for producing counterfactually augmented data that substantially improves OOD performance. We note that merely “introducing auxiliary data” is not enough to achieve the performance we observe, and the auxiliary data must be used in a disciplined manner (in our method, we are guided by causal inference methodology, i.e. matching and diff-in-diff).
> 2. Our theoretical analysis departs significantly from previous work on counterfactual data augmentation by considering sample complexity aspects instead of analyzing performance given infinite data.
> 3. The proposed method and analysis open up the possibility for using any applied causal method for generating training data. We highlight two different approaches in this paper: matching and diff-in-diff. However, in principle, other methods from the econometrics literature, such as instrumental variables and synthetic controls, can be leveraged for better OOD performance using our approach and we expect future work to significantly extend ours.
>
> **Baselines**:
> Following our model for the problem in Fig. 1, we choose baselines that are appropriate for this task since they (a) are proposed in a paper that studies this anti-causal prediction problem, and (b) achieve state-of-the-art performance on it.
>
> We have experimented with other baselines on the medical notes tasks and following your comment we will include results on GroupDRO and IRMv1 in the paper. The results can be found in the PDF attached to the rebuttal. IRMv1 underperforms the baseline from our original submission and GroupDRO is comparable, while CATO (A) has the best performance.
>
> We refrained from including these baselines in the original submission, since previous work already documented their limited success on most large scale problems (e.g. [Rosenfeld et al. 20, Kamath et al. 21, Gulrajani and Lopez-Paz 20]) and we wanted to avoid making this the focus of our work. The medical notes task is far larger in scale than datasets where these methods have shown significant gains, and devising methods that show gains in large scale problems was among our main motivations for this work. Following your comment, we see why it is important to include these results, and it led us to revise our decision. Thank you for helping us improve this aspect of the paper.
>
> **LLM-induced biases**:
> This is a great point! We have also included a discussion on this in our general response.
> Your comment about LLMs varying in their generative abilities is on point. We’ve experimented with several models in the medical notes task and will include our results on Bio-BERT, Sentence-BERT and GPT4 in our revised paper. We find that models trained on synthetically augmented data using each of these LLMs results in improved OOD performance w.r.t the baselines.
> Connecting this to your second question about biases of off-the-shelf LLMs, it is possible that LLMs introduce biases into our problem, inherited from their own training data and these biases may be specific to each LLM. This requires further study, however from our manual examination we found their quality satisfactory (see Appendix C for generation examples) and that all of them helped in OOD generalization which is our main evaluation metric (note that inaccurate counterfactual estimations in our case are only harmful if they adversely affect downstream predictive performance).
> Nonetheless, we believe that further studies are required to characterize the possible biases of LLMs in the context of our work, for instance by comparisons of the counterfactual estimation made by LLMs with a rewrite of the text made by the human whose style we are trying to emulate (over a small curated dataset).
>
> **Quality of auxiliary data**:
> This is also an important point. As in any causal matching problem, having reliable auxiliary data is crucial for selecting control examples. If the auxiliary data does not capture any information or contains wrong information we won’t be able to correctly match examples.
> Looking into our experimental analysis, we show that in two problems with radically different auxiliary data (complete patient history in the clinical narratives, and three review indicators in CeBAB) we get better performance than very strong baselines.
>
> We will add additional analysis where we corrupt the auxiliary data in both real-world experiments (we have already done that for the synthetic data).
>
> **Comments on writing**:
> Thanks for turning our attention to the typo in line 134, the term refers to the Bayes-optimal classifier, so it should be $y\in{[L]}$ (L being the number of classes) instead of $y\in{[K]}$.
> We will also move details on the LLMs we use into the main paper.
>
> **References**:
> 1. Rosenfeld et al. 20, The Risks of Invariant Risk Minimization
> 2. Kamath et al. 21, Does Invariant Risk Minimization Capture Invariance?
> 3. Gulrajani & Lopez-Paz 20, In Search of Lost Domain Generalization

---

> > ### Comment · Reviewer_8nR3 · 2023-08-11
> >
> > I appreciate the authors’ hard work in the rebuttal period. I am still concerned about the heavy reliance on LLMs in this particular application. However, I agree that the authors have provided sufficient studies addressing or acknowledging the potential imitations. As some of my concerns are resolved, I would like to increase the score by 1.

---

> > > ### Author Response · Authors · 2023-08-21
> > >
> > > Thank you very much for your engagement in the discussion and for updating your score to reflect your perception of the paper. We greatly appreciate the effort you put into the process, and hope that our added experiments, discussion on limitations of using LLMs, and other edits motivated by your review will address the remaining concerns.

---

### Official Review · Reviewer_Erys · 2023-07-07

**Soundness:** 3 good
**Presentation:** 3 good
**Contribution:** 3 good
**Rating:** 7
**Confidence:** 4

**Summary:**

The paper addresses the issue of text classifiers relying on spurious correlations, leading to poor generalization vis-a-vis out of domain, particularly in critical areas like healthcare. The authors propose using counterfactual data augmentation, guided by knowledge of causal data structure, to create text classifiers more robust to distribution shifts. Using an LLM to implement this approach, they demonstrate effectiveness in tasks such as predicting clinical diagnoses from medical narratives and observe improved out-of-distribution accuracy compared to baseline algorithms. They suggest that their method could be beneficial in dealing with various distribution-shift problems in machine learning applications. They demonstrate the utility of using language models to create counterfactuals that could improve model robustness. The paper also goes a step further in formalizing counterfactual data augmentation.

**Strengths:**

- The authors experiment with both real-world clinical diagnoses scenarios and semi-synthetic data, demonstrating the broad applicability and practicality of their proposed methodology. Working with healthcare data isn't always easy either, so credit for that choice as well.
- By using the capabilities of large language models for counterfactual generation, this work expands the potential applications of counterfactually augmented data while reducing costs associated with manually constructing counterfactuals.
- The paper offers a formalization of counterfactually augmented data, which paired with prior work allows easier understanding and replication of the process for future researchers.

**Weaknesses:**

- The method pre-supposes full knowledge of the causal structure of the data and overlooks complexity involved in real-world datasets where such information might not be readily available or accurately defined. Furthermore, assuming "no unmeasured confounding" isn't always realistic in real-world data sets especially those from healthcare, possibly limiting the model's robustness within certain contexts. Granted it is a common assumption in causal inference, but causal inference methods are also usually restricted to simple worlds that can be represented in a few variables.
- The authors acknowledge that generating versions of clinical narratives as if they had been written by different caregivers is difficult to achieve in practice. However, there's not enough discussed on substantive solutions or workarounds they employed (or considered) to meet these challenges.
- The paper needs greater discussion of prior work that has attempted to address these issues, including attempts at formalizing counterfactually augmented data and counterfactual invariance (see [1] and [2]).

[1] Divyansh Kaushik, Amrith Setlur, Eduard Hovy, and Zachary C. Lipton. "Explaining the efficacy of counterfactually augmented data." ICLR 2021.

[2] Victor Veitch, Alexander D'Amour, Steve Yadlowsky, and Jacob Eisenstein. "Counterfactual invariance to spurious correlations: Why and how to pass stress tests." NeurIPS 2021.

**Questions:**

- I think the paper could benefit a discussion of how it differs from [1] and [2]. Could you share how you view the difference (the bounds are clearly new)?
- In Section 4 you share using auxiliary data to help with counterfactual generations so the models don't hallucinate. Could you share how often did the models hallucinate (or not) when this auxiliary data was presented vs when it wasn't?
- Could you share results of a baseline where you have 2X datapoints from the original distribution (observational data alone), compared against the counterfactually augmented data (X original + X' counterfactuals)?
- In line 308, you say "across 5 runs". What is the source of randomness in these 5 runs?
- Am I correct in interpreting Figure 4's findings that seemingly more corruption leads to better performance? Could you elaborate?

**Limitations:**

The limitations are adequately discussed.

---

> ### Author Rebuttal · Authors · 2023-08-10
>
> Thank you for the in-depth review of the paper and for raising important questions that help us improve it. Please find our responses to the concerns and questions below.
>
> **Knowledge of Causal Structure**:
> While our method assumes some knowledge of causal structure (i.e. Figure 1, an anti-causal prediction problem), we clarify that it does not require a fully specified structure that includes each and every feature in the problem (e.g. every component of the auxiliary data $M$). There are multiple graphs that comply with a “no unobserved confounding” assumption given $M$, and also with the constant effect assumption (which is a functional assumption, rather than a graphical one).
> Nonetheless we agree that these are strong assumptions that require domain knowledge. In this context, it is important to note that unlike standard causal inference problems, we are interested in downstream OOD prediction abilities instead of the estimation of a specific counterfactual text (i.e. the estimation accuracy). Our generalization bound suggests that the distribution of augmented data should resemble the distribution of true counterfactuals, this can still hold when specific estimations suffer from some errors.
> Eventually, the evaluation that matters to us most is prediction accuracy compared to the baselines.
>
> **Difference from Kaushik et al. and Veitch et al.**: Thank you for raising this point, which deserved more attention in the paper. Since reviewer J1UG also asked about comparison with similar papers to the one you mentioned, we included this in our general response and elaborate here.
> The focus of our work is on studying estimation methods for Counterfactual Data Augmentation (CDA) and explaining their improved performance from the view of sample complexity. Compared to previous work on CDA, Kaushik et al. 20 (and subsequent work in Kaushik et al. 21) do not propose scalable estimation methods and rely on manual rewriting of text by humans. Veitch et al. 21 propose a regularizer that promotes conditional independence, instead of CDA. We claim and show empirically, that in large-scale problems, such methods fail due to challenging optimization and statistical complexity.  We are not aware of other works that adapt causal estimation methods for the purpose of data augmentation.
> As for formal results, Kaushik et al. 21 treat a rather specific data-generating process, and the most general treatment we are aware of is by Wang & Veitch 23 (follow up to Veitch et al. 21). These works discuss properties of CDA under infinite data, as opposed to our finite sample point of view. Specifically, our formal setting is an instance of the “purely spurious” problems defined by Wang & Veitch, where they show CDA achieves a min-max optimal model provided infinite data (as a side note, the assumptions for our estimation methods are not specific to cases where the graph in Fig. 1 holds, and hence they can be utilized in other such “purely spurious” settings). In contrast, we make an argument for the *sample complexity* of *approximate* CDA (i.e. augmentation might not be perfect).   To this end, we identify a baseline that in our problem setting is also min-max optimal provided infinite data, and draw a comparison between the sample complexities of the two methods to explain the performance we observe in practice. Finally, our main empirical contribution is favorable out-of-distribution (OOD) generalization in a complex, large-scale, real world problem of medical note classification, which further distinguishes our work.
>
> **How often models “hallucinate”**:
> Estimating this reliably warrants manual inspection of many pairs of original and generated text to determine in each instance whether the LLM filled in data that did not exist in the original text. Since our final goal is not a perfectly accurate counterfactual estimation but rather favorable downstream classification accuracy, we did not perform an extensive evaluation to determine how frequently this occurs. Instead we found examples of such incidents during our manual experimentation with the models, and verified that it is indeed a possible failure (this also makes sense considering we provide example texts with certain properties), which can be amended by matching on auxiliary data.
> Furthermore, note that because of the focus on downstream prediction accuracy, our bound implies that inaccurate estimations of the counterfactual text can damage performance, regardless of whether these accuracies are “hallucinations” or other types of mistakes (e.g. omitting data from the note). Therefore our example in section 4 is merely one instance of possible mistakes that can be prevented by additional context.
>
> **Details on experiments**:
> Following your comments, we added an experiment with equal-sized observational samples and augmented samples. Preliminary results on this new experiment, which sheds light on the effect of augmentation while controlling for sample sizes, are in the attached PDF. We should obtain the full results (across all runs and tasks) during the discussion period and will update them.
> Regarding the source of randomness across 5 runs, these correspond to different random seeds.
>
> Finally, thank you for a careful reading of the synthetic experiments. For Figure 4, lower values of $\lambda$ result in larger corruptions. Hence the 3 top curves, which correspond to CDA, show the best performance for $\lambda=0.4$ (lowest corruption in this simulation) and worst performance for $\lambda=0.2$. So as intuition suggests, and opposite to your interpretation in the review, larger corruption results in inferior performance. We think the confusing part is that a *lower* value of $\lambda$ corresponds to a *higher corruption*. We will better emphasize this in the description of the figure, thank you for turning our attention to this.
>
> **Additional references**: Wang & Veitch 23 The Causal Structure of Domain Invariant Supervised Representation Learning

---

> > ### Comment · Reviewer_Erys · 2023-08-14
> > **Thanks for the clarification**
> >
> > Thanks for the response. It clarifies many things for me. I am happy to update my score to recommend acceptance assuming the authors would be updating the camera ready to reflect these clarifications as well.

---

> > > ### Author Response · Authors · 2023-08-21
> > >
> > > Thank you very much for your engagement in the discussion, and for updating your score to reflect your perception of the paper. We greatly appreciate the effort you put into the process, and already incorporated all the clarifications in our current revision of the paper. We also completed the experiments for which the initial results can be found in the PDF, as promised.

---

### Official Review · Reviewer_J1UG · 2023-07-07

**Soundness:** 3 good
**Presentation:** 3 good
**Contribution:** 2 fair
**Rating:** 5
**Confidence:** 3

**Summary:**

This paper develops and analyzes the counterfactual data augmentation strategies for the setting where the data-generating process is anti-causal. First, they first show the Bayes optimal predictor in the setting they considered and then how can we obtain that predictor using counterfactual data augmentation. They give two different strategies to generate the counterfactual data (1) by prompting LLM and, (2) using the difference in difference method. Then they analyze the sample complexity of counterfactual data augmentation and show that it is better than that of a baseline reweighting method. Finally, they empirically show the effectiveness of their method on synthetic and real-world datasets.


**Strengths:**

**Clarity**: The paper will be well-written and easy to follow.

**Sample Complexity Analysis**: The authors show that the sample complexity of counterfactual data augmentation is better than the reweighting baseline introduced in Makar et. al.


**Weaknesses:**

1. Contribution 2 - Counterfactual Data Augmentation (CDA) as a method to deconfound target and spurious attribute: It is not clear how this observation/finding is different from the previous literature on CDA eg. Kaushik et. al. (Explaining The Efficacy of Counterfactually Augmented Data) and Joshi et. al. (An Investigation of the (In)effectiveness of Counterfactually Augmented Data).
2. Lemma 1 is similar to the claim shown in Makar et. al. (Causally motivated shortcut removal using auxiliary labels)
3. Assumption 1 (constant effect): How justified is the assumption that the spurious attribute $c$ change the previous state in an additive manner?
4. The augmentation strategy in this paper is limited to the anti-causal setting which further limits the applicability. I understand this is standard practice in the OOD generalization literature to develop a method under specific data-generating process (causal and anti-causal) but then this paper doesn’t introduce something new than the previous works.

Overall this paper seems to re-instantiate the point mentioned in previous work that argues for the effectiveness of counterfactual data augmentation. Though this paper introduces two new ways to perform data augmentation in the context of medical datasets, the overall novelty seems low.


**Questions:**

**Minor Typos/Corrections**:
1. Line 117 (Definition 1): State that $\Delta^{K-1}$ is a simplex

---

> ### Author Rebuttal · Authors · 2023-08-10
>
> Thank you for taking the time to review our paper. We greatly value your feedback and would like to address each comment you’ve provided while highlighting the essential aspects crucial for evaluating our work.
>
> **Distinction from previous work**:
> Our contribution stands out first and foremost by the study of **counterfactual estimation methods**, and also the treatment of **finite sample effects**. These distinguish our work from those of Kaushik et al. 21 and Joshi & He 22.
>
> In reference to these papers, we would like to also add Wang & Veitch 23 to our discussion. It is important to emphasize that *all the aforementioned papers do not discuss estimation methods from observed data* nor address sample complexity. The first point is extremely crucial since the type of Counterfactual Data Augmentation (CDA) proposed by Kaushik et al. 20 is infeasible in our task.
> To put this in perspective, consider letting 3000 caretakers rewrite each other’s notes enough times to obtain an adequately large training set. To perform CDA at this scale, estimation methods are necessary and we are unaware of papers that propose causal estimation methods for purposes of data augmentation.
>
> As for the finite sample analysis, your review points to this as a strength of our paper. We emphasize that this departs from previous explanations on the effectiveness of CDA. Kaushik et al. and Joshi & He discuss the effect of CDA with an infinite sample and a very specific data-generating process. Thm. 9 in Wang & Veitch gives a more general structure where CDA provably helps generalization in what they call “purely spurious” problems. In the language of Wang & Veitch, we treat is an instance of these “purely spurious” problems. In our case, there are other methods besides CDA that learn the optimal model (both methods require infinite data). Hence a finite sample analysis illuminates CDA's favorable performance w.r.t baselines (provided our CDA is of sufficient quality) instead of simply reiterating known observations.
> Thank you for this comment, we see that the point on finite samples should be better emphasized in contribution 2 instead of the other points it makes. We will rephrase it accordingly.
>
> **Lemma 1 and Makar et al.**: Footnote 2 of our paper explicitly states that this claim was proved in Makar et al. We included it in the paper using our notation since the lemma is important for motivating our method. We do not see how this is a weakness of the paper.
>
> **Constant effect assumption**: Before discussing the validity of the assumption, we clarify that its focus is on the effect being constant, rather than being additive. By definition, the quantity $x_i(c) - x_{i, pre}$ exists and can be defined as an additive effect. The strong assumption is that once we fix $c$ and the auxiliary data $m_i$, this effect is constant (i.e. does not depend on $i$). The assumption is more reasonable as we control for more auxiliary features. In experiments, we control for many factors such as prescribed medications, vitals etc. While assuming a purely constant effect is unrealistic, our generalization bound includes a distributional distance between counterfactuals and their estimations. Hence even if the assumption is imprecise, the method may still suffice to improve generalization. Weaker versions of the assumption can thus be phrased and reasoned on, yet in this work, we opt for a simple version that is easy to communicate and fit in the scope of the paper.
>
> **Augmentation limited to anti-causal case**: There is no inherent limitation to applying the estimation methods in causal prediction problems. We use the specific anti-causal problem to motivate CDA in a principled manner, as it describes the main problem we study and it is possible to analytically compare sample complexity with an alternative solution (i.e. reweighting). CDA is also appropriate in the “purely spurious” problems mentioned in response to article 1 in your review. These are more general and include some causal prediction problems. Then to obtain the counterfactual estimates, our techniques are still viable when the required assumptions hold (constant effect, or auxiliary data accounts for all unobserved factors).
> Nonetheless, we agree that like all methods for OOD generalization, ours are not effective for all possible problems and must be used with caution. Thank you for turning our attention to this, our revised version will mention the usefulness of CDA in the more general setting of Wang & Veitch 23.
>
> **The paper re-instantiates points on CDA**:
> We believe that dismissing our work as “re-instantiating the point made in previous work” on CDA is not justified for several reasons. As explained in our rebuttal, estimation methods are crucial in practice and they are applicable beyond the medical datasets we work with. Our main interest is OOD generalization, and given that there are hundreds of papers on methods for this problem, we believe that a method that significantly outperforms the baselines on a large-scale real-world task is of interest to the community. At least from our familiarity with the literature, the large majority of proposed methods do not demonstrate results of this type. We are also unaware of other works that utilize causal estimation methods for data augmentation, and our finite sample bounds are different from analyses of previous work on CDA. Overall, we think the focus of our work has a significant but small overlap with those mentioned in the review.
>
> **References**:
> 1. Wang & Veitch 23 The Causal Structure of Domain Invariant Supervised Representation Learning
> 2. Makar et al. 22 Causally motivated Shortcut Removal Using Auxiliary Labels
> 3. Kaushik et al. 20 Learning the Difference that Makes a Difference with Counterfactually-Augmented Data
> 4. Kaushik et al. 21 Explaining the Efficacy of Counterfactually Augmented Data
> 5. Joshi & He 22 An Investigation of the (In)effectiveness of Counterfactually Augmented Data

---

> > ### Comment · Reviewer_J1UG · 2023-08-16
> > **Response to Authors**
> >
> > I thank the authors for taking the time to address my questions. Also apologies for sounding dismissive of your work in the context of overall contribution and I thank you for pointing out the main contribution of the paper i.e. finite-sample complexity analysis and making CAD work in practice using causal estimation techniques. Given this, I resonate with one of the concerns raised by Reviewer 8nR3 around the reliance of this work on the LLMs and the potential bias introduced by them. I would like the authors to include the discussion they had regarding this issue in the final version of the paper. Overall, I am raising my score by 1 and would like the authors to incorporate the changes in the final version.

---

> > > ### Author Response · Authors · 2023-08-21
> > >
> > > Thank you very much for your engagement in the discussion and for updating your score to reflect your perception of the paper. We greatly appreciate the effort you put into the process, and believe that our added discussion on limitations of using LLMs will contribute to the paper and address the remaining concerns.

---

### Official Review · Reviewer_YqAy · 2023-07-10

**Soundness:** 3 good
**Presentation:** 3 good
**Contribution:** 3 good
**Rating:** 7
**Confidence:** 4

**Summary:**

The authors propose to use knowledge from the causal structure of the data to counterfactually simulate interventions on spurious features and to learn more robust classifiers. They focus on text classification tasks and argue that this approach is appropriate in prediction problems for which the label is spuriously correlated with an attribute.

Since the authors argue that their approach emulates interventions, one assumes that they referring to the second layer of Pearl's Causal Hierarchy. Here, we assume the data and the Causal Graph (CG). However, the authors left the specification of the causal graph to be addressed as future work.

Additionally, following the Pearl's Causal Hierarchy, when leading with counterfactual questions, one must assume the data and the Structural Causal Model (SCM). Given that the authors do not provide neither the CG nor the SCM, I have concerns on how their approach generates counterfactually augmented data.

**Strengths:**

This paper presents an interesting approach to counterfactual-based augmentations focusing on textual data.

**Weaknesses:**

There are some important questions to better understand the proposed approach and the validity of results.

**Questions:**

The authors assume that the writing style of caretaker C affects the vector representation of the clinical note (Fig. 1). Further, they also assume that the label Y causes both X and X^. As far as I understood, the Clinical Condition Prediction consists in, given a patient note (X), predicts the concept(s) associated with it (Y). I want to understand if this is a causal or anti-causal problem. According to Fig. 1, we pick a concept and then generates a clinical note.

In Sec. 2, the authors mentioned several approaches to invariant representation learning and counterfactual augmented data generation, however they did not select any of them as baseline(s) in the experiments section. Is it possible to provide a better understand of both (i) why the selected baselines are the most appropriate? and (ii) why the related works are not representative baselines?

Does one need to know whether the label is spuriously correlated with an attribute in advance?

Is it possible to provide details about the private held-out data used in this paper? All information necessary and sufficient to others reproduce the results using other datasets would be helpful.

**Limitations:**

I believe that the authors only mention limitations of their work in the last sentence of the Discussion section. I believe that would be interesting to improve this discussion. For instance, it is very hard to have the correct Causal Graph for most important applications, hence, how this could be an issue, and, in the absence of the causal graph, what is the best approach?

Another issue is how can we guarantee that the counterfactual instances generated during augmentation are realistic?

---

> ### Author Rebuttal · Authors · 2023-08-10
>
> We appreciate your review and questions, and are eager to clarify all points regarding our data, baselines, and assumptions.
>
> In response to questions:
> **Causal vs. anti-causal prediction problem**: As you correctly suggest in the review, we solve an anti-causal prediction problem. A true condition in the world, $Y$, along with the identity of the caretaker, $C$, determines the text we observe $X$. Recovering $Y$ is thus an anti-causal prediction problem.
> Some additional points regarding this setting: The causal model we use has several roles.
> * To provide a coarse description of the real world problem motivating this work, namely anti-causal prediction on clinical notes. We say the description is coarse since we omit (on purpose) some variables from the graph, like the auxiliary data $M$, since one can consider multiple possible graphical structures that include it where our method is adequate (we will elaborate on this in the appendix of our revision).
> * Another role of this setting is to motivate Counterfactual Data Augmentation (CDA), showing that given infinite training data it retrieves the min-max optimal solution for the problem. Note that Wang & Veitch 23 formalize a family of graphical structures they call “purely spurious”, and show CDA retrieves a min-max optimal predictor in such problems. Hence our methods are well motivated under any of these structures (which also include some causal prediction problems), and we choose the one most adequate for our problem of interest.
> * Finally, the setting allows for a formal comparison in terms of sample complexity, between approximate data augmentation and a reweighting baseline that also provably solves the problem with enough data.
>
> **Baselines**:
> 1. Following our model for the problem in Fig. 1, we choose baselines that are appropriate for this task since they (a) are proposed in a paper that studies this anti-causal prediction problem, and (b) achieve state-of-the-art performance on it.
> 2. We have experimented with other baselines on the medical notes tasks and following your comment we will include results on GroupDRO and IRMv1 in the paper. These are arguably the two most popular approaches in the literature, and the results can be found in the PDF attached to the rebuttal. IRMv1 underperforms the baseline from our original submission and GroupDRO is comparable, while CATO (A) is still the best performing method.
>
> We refrained from including these baselines in the original submission, since previous work already documented their limited success on most large scale problems (e.g. [Rosenfeld et al. 20, Kamath et al. 21, Gulrajani and Lopez-Paz 20]) and we did not want to make that the focus of our work. The medical notes task is far larger in scale than datasets where these methods have shown significant gains, and devising methods that show gains in large scale problems was among our main motivations for this work. Following your comment, we see why it is important to include these results, and it led us to revise our decision. Thank you for helping us improve this aspect of the paper.
>
> **Knowing whether at attribute is spurious**: Our setup is suitable for problems where we have some domain knowledge, and we know that the correlation between the attribute and label is spurious (whether or not a large correlation exists in training data is something we can estimate numerically). This is a standard assumption in many works on spurious correlations (Makar et al. 22, Puli et al. 22, to name a couple). Nonetheless, we think that discovery of spurious correlations is an important direction for future work.
>
> **Private held-out data**: In all clinical data experiments, we use notes from different hospitals as the held-out data. Whenever possible, we use publicly available datasets (i2b2 competitions) that have the same structure as MIMIC-III. Whenever such data is unavailable we use data from real hospitals. While we can’t publish this data for privacy reasons, it also has the same structure as MIMIC-III and i2b2.
>
> In response to limitations:
> **Knowledge of causal graphs**: We agree that to justify our methods some strong assumptions are required, as is often the case when a method involves causal estimation. However, we note that the causal graph assumed in figure 1 is rather simple to reason about, and as explained earlier CDA is also suitable for several other graphs (see Wang & Veitch 23). The strongest assumptions we make in our view are the constant effect assumption for CATO (A), and that auxiliary data M accounts for all unobserved factors for CATO (B). Following this comment, we will add a section to the appendix that describes these assumptions much more extensively and discusses the possible limitations they pose.
>
> **Guaranteeing realistic counterfactual**: While we cannot guarantee realistic counterfactuals with total confidence, we can take some measures such as letting caretakers rewrite a handful of medical notes taken by other caretakers (in the spirit of what’s proposed in Kaushik et al. 2020) and compare these to the ones generated by the LLM. The scale of this validation may be too small to certify realistic counterfactuals, but it could be valuable. In addition, it would be interesting to complement this in future work with sensitivity analysis and uncertainty estimation to gain a better sense of the possible inaccuracies of the counterfactuals.
>
> **References**:
> 1. Makar et al. 22, Causally motivated Shortcut Removal Using Auxiliary Labels
> 2. Puli et al. 22, Out-of-distribution Generalization in the Presence of Nuisance-Induced Spurious Correlations
> 3. Rosenfeld et al. 20, The Risks of Invariant Risk Minimization
> 4. Kamath et al. 21, Does Invariant Risk Minimization Capture Invariance?
> 5. Gulrajani & Lopez-Paz 20, In Search of Lost Domain Generalization
> 6. Wang & Veitch 23, The Causal Structure of Domain Invariant Supervised Representation Learning

---

> > ### Author Response · Authors · 2023-08-21
> >
> > Since the discussion period ends in a few hours and no response has been posted, we would like to summarize and slightly extend our answer.
> >
> > The main topics raised in the review are baselines, knowledge of the causal graph and verification of the counterfactuals’ quality. We have incorporated results for the required baselines in our response, showing our proposed method outperforms them and they are on-par with the previously considered baselines. For the causal graph, we emphasized that Counterfactual Data Augmentation (CDA) is well-motivated for a variety of structures (e.g. “purely spurious” cases as in Wang & Veitch 23), therefore the knowledge of the causal graph that we require is not exact and fine-grained as the reviewer may initially perceived. We focus on the anti-causal question as it well-describes our main problem of interest, and allows a formal comparison to other consistent baselines in terms of sample complexity. As for verifying the quality of the counterfactual, we mentioned an option of comparing hand-written rewrites, and would further like to emphasize that in principle, under no unobserved confounding, another option is to use validation data. That is, we can take validation data written by caretaker $c$, with auxiliary data $m$, and generate counterfactuals in the style of $c$ from notes written by other caretakers on cases that match auxiliary data $m$. Then the validation sample can be compared to the generated counterfactuals via any two-sample test.
> >
> > Further concerns included knowledge of whether a spurious correlation exists, discussion of limitations, and information of private data and. We confirm that knowledge of whether an existing correlation between the attribute and a label is spurious is required (its existence can be verified statistically), yet this is a relevant assumption in practice and it is common in the vast majority of work on the topic. Discussion of limitations will be expanded as described in other responses, and the discussion above regarding the causal graph and the knowledge it requires will also be included. Finally, we noted that our results can be reproduced from public data whenever such validation data is available .
> >
> > As your review raised several important questions, but you maintain a high certainty score, we would be very grateful for any response regarding whether our rebuttal managed to answer any of the concerns mentioned in the review. Thank you again for reviewing our paper.

---

### Official Review · Reviewer_MXTx · 2023-07-15

**Soundness:** 3 good
**Presentation:** 4 excellent
**Contribution:** 3 good
**Rating:** 8
**Confidence:** 3

**Summary:**

In this work, the authors develop causally-driven data augmentation methods to improve model robustness.

**Strengths:**

In general, I find the paper to be very, very well written and clear. The authors do a really great job of explicitly stating their assumptions, and acknowledging when certain assumptions are strong. I think this paper makes a very interesting first step in extending OOD generalization to the recent advances in LLMs!

**Weaknesses:**

I would have liked a section that describes some of the limits of using LLMs, and whether certain LLMs would be more appropriate that others. It feels like incorporating LLMs is a big part of this work, so I would have liked more context here.

I wasn't completely convinced that the extra information M would be enough to assist the model in achieving more accurate estimates, but the authors do acknowledge that this is a strong assumption.

**Questions:**

see above

**Limitations:**

The authors do acknowledge limitations of their work.

---

> ### Author Rebuttal · Authors · 2023-08-10
>
> Thank you for the insightful and positive review. We are happy to see that you think the paper is a very interesting first step in utilizing recent advances in LLMs for OOD generalization, which is exactly our goal in this work.
>
> The points for improvement that you lay out are very helpful, and we comment on them below.
>
> **Limitations of LLMs**: following this comment, we will add to the appendix a discussion on the possible limitations of LLMs for our approach. Since reviewer 8nR3 also asked about possible limitations of LLMs, we included a summary of this discussion in the general rebuttal and we repeat it below for convenience.
> * *LLM generation quality*: We acknowledge that LLMs vary in their ability to generate realistic text. It is possible that LLMs introduce biases into our problem, inherited from their own training data. This requires further study, however from our manual examination we found their quality satisfactory (see Appendix C for generation examples) and that OOD generalization also improved for models trained on the augmented data they generate. We'll include this analysis and results with different LLMs, namely Bio-BERT, Sentence-BERT and GPT4, in the revised paper.
>
> * *Counterfactual approximation*: Other than generation quality, the additional challenge in using LLMs for CDA is our ability to elicit a good approximation to the counterfactual text. Our methods rely on principles from causal inference to advance disciplined approaches for this task. While further studies are required (e.g. systematically comparing small sets of manual re-writes of texts to the elicited LLM output), we view our work as a promising first step in this direction, which we expect to be significantly extended and improved in future work.
>
> * *Effect of biases on OOD generalization*: Since we focus on OOD generalization, the limitations and possible biases mentioned above must be weighed within this context. Namely, we should bear in mind that even though generation may be biased, this bias is only harmful when it affects the generalization of a downstream classifier, and this is what we evaluate. Further, in OOD generalization we consider cases where the training data is biased in the first place, and training a standard predictive model also results in a biased solution. Hence we must weigh risks and limitations of alternative solutions vs. those of LLMs.
>
>
> **Auxiliary information may not be enough to assist a model**: As you correctly point out, we cannot be sure that the auxiliary information is enough to account for all the unobserved features relating to the writer and the text. One step we can take to further validate our method in real world settings is to obtain a small dataset where caretakers rewrite other caretakers’ existing notes in their own style (similarly to the counterfactual data augmentation suggested in Kaushik et al. 2020) and compare these against the synthetic estimates generated by the LLM. While such an evaluation would typically be of small scale, it offers another check for the validity of the approach.
>
> Another point worth mentioning on this topic is that our end goal is to obtain a classifier that generalizes well in an OOD setting. As suggested by our generalization bound in lemma 2 of the paper, this requires distributional similarity between the distribution of counterfactuals and the distribution of our estimates. So even if our estimation methods are imperfect, they may still suffice to achieve better generalization w.r.t the baselines. Understandably, this is why we view the accuracy on both ID and OOD test data as the main evaluation for our method and the baselines.
>
> Thank you again for your effort and help in improving our paper.
>
> **References**:
> 1. Kaushik et al. 2020 Learning the Difference that Makes a Difference with Counterfactually-Augmented Data

---

> > ### Comment · Reviewer_MXTx · 2023-08-10
> >
> > Thank you for your response. I've read the authors' rebuttal, and stand by my original review.

---

> > > ### Author Response · Authors · 2023-08-21
> > >
> > > Thank you very much for your engagement in the discussion and for the positive review, we greatly appreciate the effort you put into the process.

---

### Author Rebuttal · Authors · 2023-08-10

We thank the reviewers for their efforts and valuable comments, which led us to make important revisions to the paper, particularly by adding two additional baselines to our experiments (IRMv1 and GroupDRO), clarifying the connections of our work to previous papers on invariant learning and counterfactual augmentations, and adding a discussion on limitations of LLMs and possible biases they can introduce.

We are grateful for the appreciation of the novelty of our formal treatment of finite sample effects (reviewers J1UG, Erys), the superior strength of our method on real-world experiments (reviewers Erys, 8nR3), the novelty of using LLMs for counterfactual augmentation (reviewers MXTx, Erys), and for finding our paper very interesting (reviewers MXTx, YqAy, 8nR3). Moreover, we are happy to hear that all reviewers found our work to be clear and well-written.

Below we summarize our responses to the reviewers’ major concerns and review the major changes we introduced following the reviewers' comments.

1. **Additional baselines**: Reviewers YqAy, 8nR3 were curious about the performance of popular methods in OOD generalization that were excluded from our baselines. Following our model for the problem, we chose baselines that are appropriate for this task since they (a) are proposed in a paper that studies this anti-causal prediction problem, and (b) achieves state-of-the-art performance on it. Following reviewers comments, we have added experiments with two other baselines, GroupDRO and IRMv1, on the medical notes tasks. These are arguably the two most popular approaches in the literature, and the results can be found in the PDF attached to the rebuttal. As can be seen in our newly-added experiments, our method consistently outperforms these methods.

2. **Previous work on Counterfactual Data Augmentation (CDA)**: Reviewers J1UG, Erys wanted clarification about the connection of our work to previous prominent papers (e.g. Kaushik et al. 2021, Joshi & He 2022, Veitch et al. 2021). We emphasize that the focus of our work is on studying estimation methods for CDA and explaining their improved performance from the point of view of sample complexity. In comparison to previous work on CDA, these works do not propose scalable estimation methods, and we are not aware of other works that adapt causal estimation methods for the purpose of data augmentation. They also discuss properties of CDA algorithms under infinite data, as opposed to our finite sample point of view. Finally, our main empirical contribution is favorable out-of-distribution (OOD) generalization in a complex, large-scale, real world problem of medical note classification, which further distinguishes our work.

3. **Limitations of LLMs**: Reviewers MXTx, 8nR3 thought that the paper could benefit from additional discussion on the potential limitations and biases of LLMs in the context of our problem. The following points summarize a discussion that will be added to our revised paper.
* *LLM generation quality*: We acknowledge that LLMs vary in their ability to generate realistic text. It is possible that LLMs introduce biases into our problem, inherited from their own training data. This requires further study, however from our manual examination we found their quality satisfactory (see Appendix C for generation examples) and that OOD generalization also improved for models trained on the augmented data they generate. We'll include this analysis and results with different LLMs, namely Bio-BERT, Sentence-BERT and GPT4, in the revised paper.

* *Counterfactual approximation*: Other than generation quality, the additional challenge in using LLMs for CDA is our ability to elicit a good approximation to the counterfactual text. Our methods rely on principles from causal inference to advance disciplined approaches for this task. While further studies are required (e.g. systematically comparing small sets of manual re-writes of texts to the elicited LLM output), we view our work as a promising first step in this direction, which we expect to be significantly extended and improved in future work.

* *Effect of biases on OOD generalization*: Since we focus on OOD generalization, the limitations and possible biases mentioned above must be weighed within this context. Namely, we should bear in mind that even though generation may be biased, this bias is only harmful when it affects the generalization of a downstream classifier, and this is what we evaluate. Further, in OOD generalization we consider cases where the training data is biased in the first place, and training a standard predictive model also results in a biased solution. Hence we must weigh risks and limitations of alternative solutions vs. those of LLMs.

**Overview of major changes**:
1. We added two widely-used baselines (IRMv1 and GroupDRO), and an experiment with modified sample sizes (suggested by reviewer Erys). Results are in the attached PDF.
2. We made precise our connection to relevant previous works, highlighting our novelty in discussing finite sample properties of our approach and the usefulness of our method for approximating counterfactuals.
3. We added a section to discuss the possible biases that LLMs may introduce into our problem, emphasizing that further studies are required. We also note that when our goal is OOD generalization, and training data may already be biased, we need to weigh the biases of LLMs vs. alternatives.

**References**:
1. Kaushik et al. 2021 Explaining the efficacy of counterfactually augmented data.
2. Veitch et al. 2021 Counterfactual invariance to spurious correlations: Why and how to pass stress tests.
3. Joshi & He 2022 An Investigation of the (In)effectiveness of Counterfactually Augmented Data

---

### Decision · Program_Chairs · 2023-09-21

**Decision:**

Accept (poster)

**Comment:**

The paper presents an augmentation technique to improve generalization of ML models. Reviewers highlighted a concern that the method critically depends on LLMs for generating the augmentations, but otherwise acknowledged the paper's contribution, empirical experiments, and its clarity in stating necessary assumptions. I would recommend the authors to revise their paper based on the feedback, in particular, 1) discuss the limitations of relying on LLMs; 2) how this paper tackles the limitations of CF data augmentation as noted in prior work (Joshi et al.).